



# Spatially distributed snow depth, bulk density, and snow water equivalent from ground-based and airborne sensor integration at Grand Mesa, Colorado, USA

Tate G. Meehan[1,2], Ahmad Hojatimalekshah[2], Hans-Peter Marshall[2], Elias. J. Deeb[1], Shad O'Neel[1,2], Daniel McGrath[3], Ryan W. Webb[4], Randall Bonnell[3], Mark S. Raleigh[5], Christopher Hiemstra[6], Kelly Elder[7]

[1] Cold Regions Research and Engineering Laboratory; U.S. Army Corps of Engineers; Hanover, NH, USA
[2] Department of Geosciences; Boise State University; Boise, ID, USA
[3] Department of Geosciences; Colorado State University; Fort Collins, CO, USA
[4] Department of Civil and Architectural Engineering & Construction Management; University of Wyoming; Laramie, WY, USA
[5] College of Earth, Ocean, and Atmospheric Sciences; Oregon State University; Corvallis, OR, USA
[6] Geospatial Management Office; USDA Forest Service; Salt Lake City, UT, USA
[7] Rocky Mountain Research Station; USDA Forest Service; Fort Collins, CO, USA

*Correspondence to*: Tate G. Meehan (tate.g.meehan@erdc.dren.mil)

**Abstract.** Spaceborne remote sensing of snow currently enables landscape-scale snow covered area, but estimating snow mass in the mountains remains a major challenge from space. Airborne LiDAR can retrieve snow depth, and some promising results

have recently been shown from spaceborne platforms, yet density estimates are required to convert snow depth to snow water equivalent (SWE). However, the retrieval of snow bulk density remains unsolved, and limited data is available to evaluate model estimates of density in mountainous terrain. Knowledge of the spatial patterns and predictors of density is critical for accurate assessment of SWE and essential snow physics, such as energy balance and mechanics related to hazards and over-snow mobility. Toward the goal of landscape-scale retrievals of snow density, we estimated bulk density and length-scale

variability by combining ground-penetrating radar (GPR) two-way travel-time observations and airborne LiDAR snow depths collected during the mid-winter NASA SnowEx 2020 campaign at Grand Mesa, Colorado, USA. Key advancements of our approach include an automated layer picking method that leverages co- and cross-polarization coherence and distributed LiDAR–GPR inferred bulk density with machine learning. The root-mean-square error between the distributed estimates is 12 cm for depth, 27 kg/m³ for density, and 42 mm for SWE, and the median relative uncertainty in distributed SWE is 7 %.

Wind, terrain, and vegetation interactions display corroborated controls on bulk density that show model and observation agreement. The spatially continuous snow density and SWE estimated over approximately 16 km² represents the next step towards broad-scale SWE retrieval.

## 1 Introduction

Anthropogenic climate change has contributed to the declining western U.S. snowpack over the past 50 + years (e.g., Pierce

et al., 2008), and by the end of the 21st century, snow water equivalent (SWE) in the western U.S. is projected to decline by ~50 ± 10 % (Siirila-Woodburn et al., 2021). Ground observations of SWE, such as those from snow telemetry (SNOTEL) sites or manual measurements performed during snow surveys provide useful information in the context of a historical record, however with changing conditions, the relationship between these sparse observations and snow distribution patterns across



watersheds is changing. Snow varies over such short length scales that spatial interpolation from ground observations is not

possible. In response to declining snowpacks in the western U.S. (Mote et al., 2018) and the increased demand on water resources (e.g., Achhami et al., 2018), attention has been given to measuring, modelling, and developing new techniques for SWE estimates (e.g., Lettenmaier et al., 2015). The objective of NASA's snow experiment (SnowEx) campaign was to test a suite of remote sensing instruments, which measure SWE, and can be deployed on a satellite platform for global monitoring (Marshall et al., 2019). Accurate space-borne snow depth estimates have been achieved from passive microwave sensors

(Tedesco et al., 2010), Sentinel-1 radar returns (Lievens et al., 2019, 2022), WorldView stereo digital surface models (McGrath et al., 2019), and light detection and ranging (LiDAR; e.g., Deschamps-Berger et al., 2023; Hu et al., 2021) aboard ICESat-2 (Abdalati et al., 2010). LiDAR techniques have the ability to measure snow depth, by differencing repeated acquisitions during periods with and without snow cover (e.g., Deems et al., 2013), whereas active microwave retrievals are complicated by signal penetration, depolarization, and backscattering. Because of the advantages of greater spatial resolution and flexible scheduling

to target acquisitions during periods of interest, airborne LiDAR has become a prominent methodology for estimating snow depth and is being flown operationally for integration with hydrologic modelling at the catchment scale (Hedrick et al., 2018). Regardless of the choice in snow depth retrieval, an estimate of snow density is required to convert snow depths to SWE, and bulk density often provides the greatest source of uncertainty in SWE estimates, especially in deeper snow (Raleigh & Small, 2017).

Snow density is typically measured in a time consuming and spatially limited manner by excavating and weighing snow samples of a known volume from a snow pit or snow core. Because snow depth varies in space more significantly than density (e.g., Elder et al., 1991; Sturm et al., 2010) and depth measurements may be collected more rapidly, density is observed far less frequently (e.g., Elder et al., 1998; Rovansek et al., 1993). As a result, snow sampling strategies tend to be too coarse to examine the $10^0 - 10^3$ m scale spatial variability of snow density (e.g., Fassnacht et al., 2010).

Often, empirical models provide the means to spatially distribute density in SWE estimates. Linear regression models developed using snow depth alone are often unsuccessful, because the snow load only has a linear effect on bulk density while grain-bond characteristics can have an exponential effect (Sturm & Holmgren, 1998). The accuracy varies among linear snow density models that are parameterized by features such as net radiation, elevation, slope, curvature, and snow depth, as the success of such approaches is dependent on the time of year and snow climate (e.g., Elder et al., 1998; López-Moreno et al.,

2013). Successful regression models parameterized by snow depth have been split up into elevation and month of year classes (Jonas et al., 2009), accumulation and melt seasons (Hill et al., 2019), or day of year and snow cover classification (Sturm et al., 2010) to account for the effects of snow depth and snow age on density. Over timescales of days to weeks, densification processes of freshly accumulated snow result in negative correlation between snow depth and density, while over timescales of months or longer, depth and density tend to be positively correlated (McCreight & Small, 2014). By differentiating between

the short and long timescales of densification, McCreight and Small (2014) developed a linear density model capable of accurate daily density estimates for converting depth to SWE at distributed measurement stations. Hill et al. (2019) showed



improvements in accuracy compared to previous models when using sets of linear models based on snow depth and climate predictors from snow telemetry data from the western U.S., Canada, and Alaska.

Snow density can also be estimated with process-based snow models, which may account for changes in bulk snow density
due to new snowfall, metamorphism, and compaction. The representations of snow densification range in complexity, with some models utilising more simple time-dependent compaction curves and other models representing snow compaction dynamically as a function of snow viscosity and overburden pressure. Essery et al. (2013) found that the dynamic models offer more consistent and accurate characterizations of snowpacks. However, there has been a range of performance in snow density simulations, even for a single physics-based model. For example, Snobal (Marks et al., 1992) yielded low errors (mean absolute
difference of 24 kg/m$^3$) in a study of the California Sierra Nevada (Painter et al., 2016) but higher errors (root-mean-square error up to 142 kg/m$^3$) in a study of the Canadian Rockies (Lv & Pomeroy, 2020). Egli et al. (2009) found similar capabilities in estimating snow density with physics-based models and empirical models at a point location in Switzerland. In contrast, Raleigh and Small (2017) found that the choice of snow density model (empirical or physical) produced differences in spatial distributions and basin mean estimates of snow density in California, and one of their main conclusions was that the major
source of uncertainty in SWE estimates from airborne lidar was due to the accuracy of the density estimate used.

Despite numerous techniques for modelling snow density, there are few studies comparing how they characterise spatial variations in snow density and the underlying processes related to density variations, largely due to limited density datasets. Spatial validation has been limited because few techniques exist for continuous spatial measurement of snow density, and the limited spatial support of in situ observations proves too intensive for analyses of spatial variability in snow density.
Relationships between dielectric permittivity (a main constituent of electromagnetic wave propagation) and snow density (e.g., Matzler, 1996), provide radar remote sensing an advantage for snow density retrieval. Passive microwave emission measurements combined with radiative transfer modelling is an established theoretical basis for retrieving snow density and ground permittivity (Schwank et al., 2015). Though this technique has been proven experimentally at the plot scale (Lemmetyinen et al., 2016), initialising the ground temperature and roughness remains challenging for satellite application.
Ground-penetrating radar (GPR) records the amplitude and travel-time of a series of echoes from short-pulse electromagnetic waves as an image in range-time and position coordinates. With information about the propagation velocity of the snow, GPR analysis can estimate the snow depth, or by exploiting a ray path function of travel-time versus antenna separation (offset) the velocity can be estimated, and thereby the snow depth and density. Interpreting GPR transect imagery is a laborious process that requires expert knowledge of digital signal filtering and manual picking of travel-times. Though intricate, multi-offset
GPR analysis has continuously measured snow depth and density along transects of hundreds of metres in alpine mountains (Griessinger et al., 2018) to tens of kilometres in Greenland (Meehan et al., 2021). By combining drone-based aerial photogrammetry with GPR travel-times, snow density has been estimated along 100 m scale transects, which have been analysed for densification time-series (McGrath et al., 2022; Valence et al., 2022) and spatial interpolation across the study-plot scale (Yildiz et al., 2021). Radar derived snow observations require calibration and validation, which remains limited due



to the challenges of in situ sampling. Further hindering comparisons, in situ snow observations exhibit shorter length-scale variability than can be inferred with radar techniques, which tend to reveal lower spatial-frequency patterns.

Here, we evaluated whether combining GPR two-way travel-time (TWT) observations with airborne LiDAR snow depths can accurately estimate density, and then be used to derive spatially continuous estimates of bulk snow density (and thus SWE), and whether these estimates reveal repeatable and physically realistic spatial patterns at Grand Mesa, Colorado, USA. We developed a novel approach for automatically interpreting TWTs using co- and cross-polarization coherence of GPR cross-section images and implemented a fast decision tree search to pair co-located GPR and LiDAR observations. Using variogram analysis, we determined the spatial length scales of variability in LiDAR snow depth, GPR TWT, the resulting bulk density, and SWE, to quantify differences in the snow distribution between open and forested areas. Snow density inferred from the integration of GPR and LiDAR information was then distributed at 1 m resolution over a ~16 km$^2$ study domain using machine learning regression techniques trained on LiDAR features. Spatial covariance derived from the snow density variogram was developed into a baseline, statistically distributed snow density model representative of the mean and spread of density observed in snow pits, using random normal field theory. We assessed the random field model and snow density models distributed via linear (MLR), random forest (RF), and artificial neural network (ANN) regression for accuracy, structural similarity, and ability to represent wind shelter and exposure. We evaluated the degree of improvement in SWE at the sub-catchment scale by comparing distributed SWE estimates from regression techniques and random-field synthesis to those estimated from mean density measured in snow pits with the airborne LiDAR snow depths. We set 10 % average uncertainty as a benchmark for our approach (National Academies of Sciences, Engineering, and Medicine, 2018), and identified areas where uncertainties exceed this standard.

Our work addresses the need for high accuracy, distributed density measurements to improve parameterizations of snow densification processes and reduce model uncertainty. The measurement and spatial distribution of snow density will increase the snow-water resources community awareness of the spatial variability in snow density and its importance. Knowledge of the spatial patterns and predictors of density is applicable for radar remote sensing retrievals of snow, essential snow physics such as mechanics related to hazards and over snow mobility, and thermal conductivity as it pertains to the energy balance and modelled snowpack evolution.

## 2 Methods

### 2.1 Study area

Grand Mesa, Colorado, is a high-elevation subalpine plateau with an average elevation of ~3,200 m and an area of ~1,300 km$^2$. Grand Mesa has a cold and dry continental snow climate, low relief, and varying vegetation cover from shrub steppe and subalpine meadow to dense conifer forest. These factors, along with the proximity to a regional airport make Grand Mesa a near-ideal study area for evaluating airborne snow remote-sensing techniques and developing many challenging snow remote-sensing advancements, such as our density retrieval.



The Grand Mesa NASA SnowEx Intensive Observation Period (IOP) spanned 27 January – 12 February 2020. During that time, more than 150 snow pits were excavated and nearly 38,000 in situ snow depth measurements were collected. The snow pits were distributed in forested and open areas along the swaths of the three airborne remote-sensing campaign flight lines
(Fig. 1).

## 2.2 GPR data acquisition

Two GPR instruments were operated during the first week of the Grand Mesa IOP. In the forested areas, a conventional L-band GPR was pulled by ski in forested areas of central Grand Mesa and into open areas on the forest perimeters during 30 January – 1 February and 5 February (Webb, 2021), which was equipped with a Global Positioning Satellite (GPS) receiver
with 2.5 m horizontal accuracy. Whereas, in open areas, we deployed multi-polarization L-band GPR fastened within a sled that was pulled by snowmobile at approximately 3 m/s in the open areas of the central and south regions of western Grand Mesa on 28 and 29 January, and 4 February 2020 (Meehan, 2021). The snowmobile was driven along the edges of the many forested stands in the survey domain but did not travel through densely treed areas. The multichannel L-band GPR configured with one transmitting antenna and two receiving antennas that were oriented parallel (H) and orthogonal (V) to the transmitter
(H). The transmit and receive antennas were separated by 25 cm. Using this GPR configuration we simultaneously acquired the radar imagery in co- and cross-polarizations (HH & HV). A Global Navigation Satellite Systems (GNSS) receiver with sub-meter accuracy was located on the snowmobile 5 m away from the GPR array, so we applied a geometric correction to relocate the coordinate positions to the antenna midpoint of each channel.

The GPR systems were operated continuously, collecting approximately 30 traces per second, given the duration of the time
window for each trace (30 ns), the sample interval (0.1 ns), and the number of stacks acquired (2). Due to differences in the travel speed, the spatial interval of the GPR traces collected via snowmobile is approximately $10 \pm 1$ cm, while the interval for traces collected by ski is $5 \pm 1$ cm. We used piecewise cubic Hermite interpolating polynomials (Kahaner et al., 1989) to fix a geolocation to every acquired trace, as the GPS acquisition rate was 1 Hz. For the multi-polarization GPR, we estimated the accuracy of the georeferencing at 70 cm, which is on the order of the GPR footprint. This estimate follows from adding the
horizontal dilution of precision (50 cm) and uncertainty in the sled location (50 cm) in quadrature. Throughout this week, we acquired 144 km of quasi-gridded and spiralled snowmobile-driven radar transects, and 16 km of skied spiral transects in the forest. Spiral transects were coincident with depth measurements. We used a 4.5 km by 3.5 km portion of the snow-on LiDAR acquisition to bound the GPR transects (Fig. 1) and omitted any transects acquired beyond the LiDAR boundary.

## 2.3 GPR data processing

Multi-polarization radargrams were processed using the following automated routine. We applied a frequency-wavenumber (F-K) filter as a 2D band-pass filter (Kim et al., 2007). Time-zero correction was performed automatically using the Modified Energy Ratio first break picker (Wong et al., 2009). We removed coherent noise by subtracting the median trace from the radargrams (Kim et al., 2007). The trace amplitudes were corrected for spherical divergence by applying t-squared scaling as



a signal gain function (Yilmaz, 2001). In a step of random noise removal, we then applied edge preserving smoothing
(Kuwahara et al., 1976). This routine emphasises the continuity and amplitude of the ground reflection, which benefits the
method for automatically picking the travel-times. The GPR data within forests were manually interpreted and picked using a
semiautomatic algorithm and are available through the National Snow and Ice Data Center (Webb, 2021). The slower paced
data acquisition by ski improves the quality of the radargram, which benefits the tracking of the ground surface in the more
variable forest environment.

### 2.3.1 Multi-polarization coherence for automatic two-way travel-time determination

The rough ground depolarized the L-Band radar signal and thus we used the coherence between the co- and cross-polarized
channels as a filter that illuminates the ground reflections and removes the planar reflections of the snow stratigraphy. We
paired the co- and cross-polarization radargrams into shot gathers, which are the bins of traces that share the same transmitter
location. The automatic travel-time pick is determined by maximising the coherence between the co- and cross-polarization
shot gathers. For each pair of traces, we applied the unnormalized cross-correlation sum to measure the coherence,

$$C(t) = \frac{1}{2} \sum_{j=1}^{N} \left\{ \left[ \sum_{i=1}^{M} S_{i,t} \right]^2 - \sum_{i=1}^{M} S_{i,t}^2 \right\}, \tag{1}$$

which is half of the summed difference between the energy of the stacked traces and the energy of the input traces (Neidell &
Taner, 1971). The calculation in Equation 1 is performed in a sliding window over $N = 11$ samples that is evaluated at every
sample ($t$) of the GPR signal ($S_{i,t}$) for channels $i$ ($M = 2$). The HH-HV coherence ($C_{HH-HV}$) at each shot location is then
normalised by the maximum coherence, ($C$):

$$C_{HH-HV} = \frac{C}{(C)} . \tag{2}$$

Small (one-wavelength) offsets introduce waves that have approximately normal incidence to the reflection horizons, such that
nonlinear effects of travel-time moveout are negligible and snow depth can be directly inferred from the measurement of TWT.
Because the offsets are equal, the travel-times to the ground for each channel are equal within a small error (due to variability
of the ground surface inside the footprint), and therefore the two channels sum coherently.

We automatically chose the travel-time with the maximum coherence of each trace and subtracted 1 ns (1/2 wavelet) to estimate
the first break of the reflection (Booth et al., 2010). We then applied a median filter to remove outliers and reviewed the
automatic picks for any systematic errors, less than 1 % of the automatic picks requiring manual correction. To illustrate this,
automated picks are overlaid on the radargrams of a 900-metre-long transect in Fig. 2. The resulting TWT data produced from
this method and used in this study are available through the National Snow and Ice Data Center (Meehan, 2021).



### 2.4 Snow observations

#### 2.4.1 In situ measurements

Snow pit observations and manual depth probe measurements were collected throughout the IOP to serve as validation for SWE and snow depth retrieved by airborne remote sensing. Snow pits were measured for the snow depth, density, water equivalent, temperature, wetness, liquid water content, grain size, and stratigraphy (Vuyovich et al., 2021). Snow density ($\rho_{s,pit}$) was measured continuously every 10 cm from the snow surface to the ground using a 1000 cm³ wedge sampler, with duplicate samples. If the difference between the two measurements at a given depth exceeded 10 %, the density was sampled a third time, and bulk density was then calculated by averaging all measurements for each snow pit. The average density sampled from each of the columns shows high repeatability with a root-mean-squared-difference of 2.5 %. By regressing average density against the time elapsed throughout the campaign, we observed linear trends in snow densification: 0.07 kg/m³/hr in forested areas and nearly double in open areas (0.13 kg/m³/hr). Because the density snapshot we retrieved is valid for the time of the LiDAR flight, we corrected measured density to 12:00 PM on 1 February using these rates. Liquid water content was estimated by combining the density and in situ measurements of dielectric permittivity in an empirical formula, which showed that the snowpack remained almost completely dry throughout the IOP (Webb et al., 2021). Snow depth measurements ($h_{s,probe}$) were collected using geolocated probes (± 3 m spatial accuracy) along spiral transects (~60 m radius) centred around pits (Hiemstra et al., 2020).

#### 2.4.2 LiDAR snow depth

Snow depth ($H_{s,LiDAR}$) was estimated from repeated airborne LiDAR point cloud surface elevations of snow-free and snow-covered terrain using the Multiscale Model to Model Cloud Compare (M3C2) method (Lague et al., 2013). The M3C2 method operates directly on point cloud data, computes the local distance between two point clouds at a scale that is appropriate for the surface roughness, and estimates a confidence interval for each distance measurement. The Airborne Snow Observatory (ASO) performed the snow-free acquisition on 26 September 2016 (Painter et al., 2016; Painter & Bormann, 2020), and NV5 Geospatial acquired a time-series of snow-covered surface elevations during the IOP, both with a point density of approximately 20 points/m². We selected the 1 February 2020 flight to minimise temporal differences with the GPR and resulting errors due to snow redistribution and densification. We removed vegetation following methods in Štroner et al. (2021) and transformed the 2016 snow-free vertical datum into NAVD88/Geoid 12B (the same as 2020 snow-on) using NOAA VDatum 4.3 software (NOAA, 2021). Then, we applied the M3C2 method to estimate snow depth on a 1 m grid. The relative accuracy of the snow depth measurement was estimated at 7 cm, based on the maximum standard deviation of the M3C2 method, which agrees well with previous lidar error assessments (Hojatimalekshah et al., 2021). After computing the snow depth, the 3 m ASO bare-earth and vegetation data products were resampled to the 1 m resolution of the snow-covered SnowEx 2020 LiDAR acquisitions and the coordinate system was transformed from UTM zone 13 N to UTM zone 12 N.





### 2.4.3 LiDAR – GPR estimated density

We combined the LiDAR snow depths with the GPR TWTs to calculate the radar wave velocity, which in dry snow is only a function of density. We applied a k-d tree searcher (Bentley, 1975) to co-register the LiDAR coordinates within a 1 m radius

of the GPR TWTs. We then used the median values of the TWTs within a 1 m radius of these coordinates to interpolate to the LiDAR grid, and the estimated velocities were inverted for density. By calculating the maximum cross-correlation lag on continuous segments of transect data, we determined that the overall accuracy of the spatial registration between the LiDAR and GPR varies on the order of a few metres. We found that errors in the co-registration of these data are the leading source of error in the estimated densities.

The average electromagnetic wave speed of the snowpack was estimated using

$$v_{s,LiDAR-GPR} = 2\frac{H_{s,LiDAR}}{\tau} \; , \tag{3}$$

for each of the coincident LiDAR snow depths ($H_{s,LiDAR}$) and GPR two-way travel-times ($\tau$). We then related the electromagnetic wave speed to the dry snow density using the Complex Refractive Index Method (CRIM; Wharton et al., 1980)

$$\rho_{s,LiDAR-GPR} = \rho_i \left( 1 - \frac{v_a(v_i - v_{s,LiDAR-GPR})}{v_{s,LiDAR-GPR}(v_i - v_a)} \right) . \tag{4}$$

The CRIM equation relies on the known wave speeds of the pore-space ($v_a = 0.3$ m/s) and ice matrix ($v_i = 0.169$ m/ns), the measured bulk wave speed of the snowpack ($v_s$; Equation 3), and the density of ice ($\rho_i = 917$ kg/m$^3$) to determine the dry snow density ($\rho_{s,LiDAR-GPR}$; Equation 4).

### 2.4.3.1 Error analysis of LiDAR – GPR estimated density

We conducted a sensitivity analysis to evaluate how the errors in radar travel-time and LiDAR snow depth affect the estimated snow density. This involved establishing a level curve through the average snow values of the study area (mean density 276 kg/m$^3$, mean TWT 8 ns, and mean snow depth 96 cm) and applying perturbations to evaluate the density error (Fig. 3). Perturbations of up to ± 1 ns were added to the TWT and ± 15 cm were added to the depth. After the TWT and depth perturbations were applied, the densities were evaluated, and the mean (276 kg/m$^3$) was subtracted from this result to measure

the density perturbations. The error bars of Fig. 3 represent the LiDAR root-mean-square difference (RMSD) evaluated by co-located depth probing (11 cm) and at the GPR TWT crossovers (0.9 ns). At the 1-sigma level, errors of approximately ± 150 kg/m$^3$ can be expected from this sensor integration method. Using variogram analysis (Section 2.5), we determined that beyond 10 m these errors are spatially uncorrelated suggesting that the errors are random and can be treated with filtering.

The combined measurement and geolocation errors in LiDAR-derived snow depths and GPR TWTs may translate to errors in the retrieved density that are larger than the range of densities observed in the snow pits. To extract a meaningful density signal, we reduced the random error to ± 30 kg/m$^3$ by filtering outliers. We chose the interquartile range of the LiDAR – GPR inferred densities as the threshold for determining outliers, because the 25$^{th}$ and 75$^{th}$ percentiles envelop the range of snow



densities observed in the snow pits. We then applied a 2D median window with a 12.5 m radius (chosen to extend beyond the correlation length of the errors) to smooth the densities and interpolate those at the outlying locations.

Although we only included GPR observations within 100 hours of the 1 February LiDAR flight, seeking a potential bias related to effects of densification and redistribution, we regressed $\rho_{s,LiDAR-GPR}$ against hours elapsed prior to and after 1 February. Separate linear trends were identified in the forested and open regions traversed with the GPR. As conducted for the snow pit density observations, the trends were centred about February 1, and removed from the inferred density observations.

### 2.4.4 Wind and terrain exposure

### 2.4.4.1 Meteorological station wind data and snow transport

As part of the SnowEx campaigns at Grand Mesa, five meteorological stations were installed between 2016 and 2017 and operated through the 2021 water year (Houser et al., 2022). Of these sites, 3 m elevation wind speed and direction data measured at Mesa Middle (MM) and Mesa West (MW) were examined as validation for snow transport potential and to quantify differences in exposed and sheltered terrain. The MW station was in exposed western terrain of the mesa, 350 m west of the study domain boundary. The MM station is sheltered within a dense stand of conifer trees 18.7 km east of the study domain boundary.

Wind data were examined from 1 October 2019 through the end of the SnowEx IOP on 12 February 2020. The MW station wind rose is presented in the Appendix C (Fig. C.1a). Hourly air temperature data parameterized the threshold for snow-transportable wind speed (Li and Pomeroy, 1997). For values exceeding this threshold, minus the 95 % confidence interval, we then determined the median wind speed and direction for snow transport as 6.4 m/s at 200 degrees (Fig. C.1b).

### 2.4.4.2 Maximum upwind slope and wind factor parameters

Serving as an additional comparison, we computed two parameters (maximum upwind slope and wind factor parameters; Winstral et al., 2002) from the 1 February lidar-derived snow surface elevations. The maximum upwind slope characterises the degree of wind exposure for a given pixel. A wind sheltered pixel has positive slope, indicating that higher terrain exists upwind of the pixel. Conversely, wind exposed terrain has a negative slope value, with lower elevation terrain in the upwind direction. The maximum upwind slope parameter (Sx) was calculated for each azimuth from 0 to 355 degrees in 5-degree increments, averaged over ± 15-degree overlapping bins (Winstral et al., 2002). A search distance of 25 m was applied in the calculation of local Sx, while a 250 m distance was used to calculate the outlier length scale Sx between 25 and 250 m from the pixel. The local and outlying Sx parameters were differenced to calculate the slope break parameter, Sb (Winstral et al., 2002).

The wind factor parameter determines the degree of wind exposure or shelter for a given pixel, based on Sx, Sb, vegetation proximity, and the average scalar multiple by which wintertime winds at the MW station exceed those at the MM station (Winstral et al., 2002). The value 2.18 determined from the wind speed data at MW and MM, is in close agreement with the value of 2.3 determined by Winstral et al. (2002) and was applied to compute the wind factor by inversely rescaling Sx to the range between 1 and 2.18, where the larger values indicate more wind exposure. Vegetated, wind-sheltered zones were



identified as pixels within a 3 m buffer of LiDAR vegetation greater than 0.5 m height. The wind factors of sheltered vegetated areas and regions of Sb which exceeded the 97.5th percentile were arbitrarily reduced by 10 % to enhance the effective wind shelter provided by vegetation.

Rather than incorporating the Sx and wind factor parameters as model predictors of snow density, we utilised this information as explanation of the patterns and processes captured by the machine-learned regression ensemble. To verify the GPR-LiDAR estimated training data and the model results, we calculated the correlation between the model input and output and the Sx and wind factor rasters for all wind directions. Because pixels with high wind exposure have negative Sx values, and we hypothesise that wind exposed terrain will have greater snow density (e.g., due to formation of wind slabs), the wind direction 295 expressing the largest negative correlation was recognized as the best explanation of density patterns determined by Sx. Because the wind factor was inversely rescaled, the wind direction expressing the largest positive correlation with the snow density was recognized as the best explanation of density patterns determined by the wind factor parameter. Agreement between the prevailing wind direction responsible for snow transport measured at the MW station and the wind direction expressing the strongest correlation with model inputs and outputs, we suggest, would bolster the method of solving for snow 300 density by integrating GPR and LiDAR information and support the spatial patterns predicted by regression model outputs, which were not trained with wind information.

## 2.5 Spatial scales of variability for snow depth, travel-time, density, and SWE

We examined the differences in snow properties between forested and open areas using generalised relative semi-variograms (Isaaks & Srivastava, 1989). The generalised relative semi-variogram describes the average percent variability, relative to the 305 mean, as a function of separation distance between observations. To estimate the spatial variability of the snow depth, TWT, density, and the resulting SWE of the 1 m gridded data along the radar transects, the experimental variograms were first calculated in 1 m bins up to a 250 m lag, and then fitted with exponential models via least-squares to estimate the range, sill, and nugget parameters (e.g., Cressie, 1985). We used an exponential variogram model, for which the correlation length is equal to three times the range parameter. We created 250 realisations of the experimental variogram calculation using Monte 310 Carlo simulation with 10 % random subsampling to assess the mean and standard deviation of the variogram parameters (Efron & Tibshirani, 1986).

To serve as a baseline for model assessment, a snow density model was synthesised from the statistics of in situ density measurements and the correlation length of snow density estimated via variogram analysis. Provided the empirical variogram function, a covariance matrix was determined between all pairs of points in the ~16 km² domain. Using Cholesky 315 decomposition, the large covariance matrix was efficiently inverted to determine a matrix of weights with the desired covariance properties (Vecherin et al., 2022). The synthetic snow density model was then generated by multiplying a normal random vector with zero mean and standard deviation of the in situ observations by the weighting matrix and adding the mean value of the density observations.



### 2.6 Regression modelling of spatial snow density

To distribute the spatial observations of average snow density to areas without GPR observations, we tested three regression techniques: Multiple Linear Regression (MLR; Andrews, 1974; Appendix A.1.1), Random Forest Regression (RF; Breiman (2001); Appendix A.1.2), and Artificial Neural Network Regression (ANN; Jain et al., (1996); Appendix A.1.2). We examined the ~16 km$^2$ area of the LiDAR domain, which closely bounded the extent of the GPR survey. A set of normalised predictor variables, notated with capital lettering, were developed using the elevations of four LiDAR rasters: bare earth elevation ($Z_g$),
snow-covered elevation ($Z_s$), snow depth ($H_s$), and vegetation height ($H_{veg}$); the aspect, slope, X and Y derivatives of the elevation rasters (excluding $H_{veg}$); and the distance to the nearest vegetation $\geq 0.5$ m ($S_{veg}$). Aspect rasters were transformed by the cosine to remove wrapping ambiguity around north. We smoothed the elevation, vegetation height, and snow depth rasters using a median filter with a 5 m x 5 m window, and the derivatives of these rasters (slope, aspect, dx, and dy) with a 25 m x 25 m window. Regression models were trained on the LiDAR–GPR estimated snow density using cross-validation and
were applied to the surrounding terrain. For detail on the parameter estimation and predictor importance see Appendix A.

### 3 Results

#### 3.1 Snow depth

The LiDAR-derived snow depths show an overall trend increasing from west to east, in addition to smaller scale patterns near vegetation, with deeper snow around the perimeter of treed areas and shallow snow on the ground beneath tree canopies (Fig.
4). This pattern is consistent with previous snow depth distribution studies of Grand Mesa (e.g., McGrath et al., 2019). The mean LiDAR snow depth for the entire domain is 92 cm with a standard deviation of 18 cm. In open areas ($H_{veg} < 0.5$ m), the mean LiDAR snow depth is $96 \pm 15$ cm, while in the forest ($H_{veg} \geq 0.5$ m), the mean LiDAR snow depth is $79 \pm 23$ cm.

The in situ snow depth observations ($h_{s,Pit}$ and $h_{s,Probe}$) compare well with the LiDAR snow depths throughout the entire domain ($R^2 = 0.61$, RMSE = 11 cm, ME = 0 cm), however within the open and forested domains individually, LiDAR and
GPR estimated snow depths are uncorrelated with in situ snow depths (Table 1).

#### 3.2 LiDAR – GPR estimated density

The LiDAR–GPR inferred average snow density shows repeatable structure at the many crossover locations and greater variability in the open terrain than areas sheltered by forest canopies (Fig. 5). The integrated LiDAR and GPR data resolve lower spatial frequency patterns than the snow pit observations, which are sparse and have limited spatial support. When
compared to snow pit observations, the relative RMSE among the 37 snow pits that are within 12.5 m of the GPR transects is 13 % (Table 2). The GPR data were acquired within a few metres of, but not directly beside the faces of snow pits, which necessitated a radius for pairing observations. The choice of 12.5 m matches that of the filtering during the sensor integration





step (Section 2.4.3). Measurements accumulated over 12.5 m distance introduce inherent variability on the order of 10 % (Section 3.3), which, along with differences between representative observation scales may explain the weak correlation between estimated density and the in situ measurements. The standard deviation of $\rho_{s,Pit}$ encompasses the mean of $\rho_{s,LiDAR-GPR}$ and vice versa, and the data are both normally distributed as evidenced by Z-test (Appendix B). The maximum upwind slope and wind factor parameters evaluated on GPR transects show the strongest correlation (R = -0.45 and R = 0.48, respectively) in the directions 225 and 220 degrees (see Table C.1). Within the width of bins used to calculate the wind parameters, the optimal wind direction uniquely agrees with the prevailing winds able to transport snow and suggests that the method retrieves density patterns which are related to the degree of exposure and shelter due to topography and vegetation.

**3.3 Spatial variability of LiDAR snow depth, GPR travel-time, density and SWE**

The generalised relative semi-variogram allows us to examine the expected percent variability between observations at a given distance separation and the differences in the length scales of variability among depth, density, SWE, and TWT within the forested and open areas of Grand Mesa (Fig. 6). Table 3 overviews the generalised relative semi-variogram parameter estimates (nugget, sill, and correlation length). The variability in depth, SWE, and TWT are lower, while variability in density is greater within the open areas than within forest stands and the length scale of variability is longer. The larger (roughly 10 m) spatial support of the LiDAR – GPR estimated densities cannot directly sense subpixel correlation lengths and potentially missed a zero-to-five-metre scale-break that is more comparable to the spatial support of in situ density observations. However, we find that the expected variability among co-located $\rho_{s,LiDAR-GPR}$ is approximately 2 %, which is consistent with replicated in situ density observations ($\rho_{s,Pit}$) having approximately 2.5 % variability on average. Compared to the estimated correlation length of density (~100 m), the median distance between snow pits is ~150 m, which indicates that average snow pit observations are independent of each other and unable to resolve spatial patterns. Supporting the basis that in dry snow, depth and density formulate TWT and SWE, the relative length-scales of variability for SWE closely resemble that of TWT and indicate that TWT is a better informer of SWE than either depth or density, independently.

**3.4 Learned Regression and Random Field Modelled Density**

Using supervised learning techniques three models were generated from LiDAR information (Fig. 7 a, b, and c). Using prior information from the in situ snow pit observations a randomly distributed density was synthesised (Fig. 7 d). The mean of the regression-based ensemble was taken to generate $\rho_{s,\overline{Ens}}$ (not shown). Generally, the regression models predict higher snow density in the open and exposed areas than in areas that are protected from the wind by trees. Each of these five models are evaluated against $\rho_{s,LiDAR-GPR}$ and $\rho_{s,Pit}$ (Table 2). As discussed in Appendix A.1.2, the model hyperparameters were developed such that the variance of the predictions in pixels where training data exists matches that of the predictions. This coincided with an R² of approximately 0.8. Evaluation against snow pit density does not depict an obvious best model choice. However, we have selected the average of the regression models, as the random field contains little meaningful spatial



information, and the average of the ensemble spreads the strengths of each regression model (i.e., accuracy of training data and representativeness of wind-induced densification). The spatial similarity of these models is presented in Appendix A.3.

**3.5 Model representation of wind, terrain, and vegetation effects**

Snow density inferred by the LiDAR – GPR retrieval method and modelled using regression approaches displays spatial patterns that we hypothesise are representative of the interactions between small-scale topography, vegetation, and wind. To test this hypothesis, we explored the correlation between the maximum upwind slope and wind factor parameters developed by Winstral et al. (2002). For each wind direction, the three learned regression models and the ensemble mean were correlated to the wind exposure parameters (Fig. 8). Based on the maximum correlation observed for the wind factor, and the minimum observed for the upwind slope, the patterns of snow density agree with the prevailing south-southwest wind direction (Appendix C). No correlation was evident between these wind exposure parameters and the randomly distributed snow densities (Table C.1). A larger correlation is observed for the wind factor than maximum upwind slope, which suggests that the role of wind shelter by vegetation, and not only terrain, has an important quantifiable effect on snow density. The lack of a large-scale topographic trend in density, such as one driven by elevation or aspect, evinces the role of forest vegetation on density.

**3.6 Spatially distributed snow water equivalent**

SWE was distributed within the ~16 km$^2$ domain by multiplying the mean snow density of the regression ensemble estimated with the LiDAR snow depths (Fig. 9). Decreased SWE and increased variability within forest stands corroborates previous work on wind-terrain-vegetation characterization of Grand Mesa (Webb et al., 2020). Wind redistribution is also evidenced by snow drifts which tend to have less SWE on the windward side and increased SWE on the leeward side. Metre-scale stippling patterns are the effect of low-stature vegetation ($H_{veg}$ < 0.5 m) and boulders, which tend to reduce snow depth and to a lesser degree reduce the modelled average density. The averages and standard deviations of SWE measured in snow pits and distributed throughout the study area domain by combining either the average snow pit density (276 kg/m$^3$) or the ensemble modelled density with LiDAR snow depths is presented in Table 4. It is unclear which method best represents the true water equivalent for the study domain because the true SWE distribution is not observed.

We compared the SWE distributed using LiDAR snow depths multiplied by the average density in each respective domain measured from snow pits, the regression modelled mean ensemble densities, and the random field densities to observations at all snow pits within the study domain (Table 4). SWE distributed from a single average density has lower bias (by 9 kg/m$^3$) and root-mean-squared-error (by 4 kg/m$^3$) than $b_{s,LiDAR-\overline{Ens}}$. The ensemble-modelled densities explain more variation in the distributed SWE estimates than those distributed randomly with an appropriate correlation length and prior mean and spread but maintain a larger bias. Using an average measured value performs slightly better, which suggests that depth is primary to SWE in this environment, yet covariance exists between snow depth and regression modelled density. Assessed at the average value of SWE for all 96 snow pits, the SWE distributed using $H_{s,LiDAR}$ and $\rho_{s,\overline{Ens}}$ has a negative bias of approximately 7 %,





which is within the uncertainty of SWE distributed from LiDAR snow depths and the mean of snow pit density measurements (Appendix B.2).

**3.7 Contributions of SWE uncertainty**

The errors between $h_{s,LiDAR}$ and $h_{s,Probe}$ are positively correlated when compared to $h_{s,LiDAR}$ ($R^2 = 0.16$), as are the errors between 415    $\rho_{s,\overline{Ens}}$ and $\rho_{s,Pit}$ are when compared to $\rho_{s,\overline{Ens}}$ ($R^2 = 0.35$), but the errors among snow depth and density are uncorrelated with negligible covariance. Using simple linear regression, we modelled the errors as a function of the LiDAR measured snow depth or the regression ensemble mean density. Following from the propagation of errors for relative errors in snow depth and density, we estimated the SWE uncertainty to first order (Raleigh & Small, 2017). The distributed relative SWE uncertainty is presented in Fig. 10 and is negatively correlated with the distributed SWE ($R^2 = 0.44$). The median SWE uncertainty is 7.6 %, 420    which breaks down to 10.6 % median uncertainty in the forest and 6.9 % median SWE uncertainty in the open areas. Snow depth and density had approximately equal contributions to SWE uncertainty at 4 %. Within the open areas LiDAR snow depth contributes 3 % uncertainty, while in the forested areas snow depth contributes 8 % median uncertainty. Snow density uncertainty is 4 % in both open and forested areas.

**4. Discussion**

This work advances the utility of GPR for seasonal snow applications, by resolving spatial snow density and SWE through the integration of remotely sensed LiDAR and GPR observations. These derived data and the validation data acquired during the NASA SnowEx 2020 IOP at Grand Mesa, Colorado, USA, provide a core set of observables that can better inform essential snow research. Grand Mesa is a good site for testing our approach of combining LiDAR and GPR for density and SWE retrieval yet presents many challenges for GPR analysis because of the abrupt discontinuities along reflection horizons due to vegetation 430    and boulders on the ground surface. By exploring effects of depolarization on L-Band GPR signals, we developed a new, automated GPR processing workflow that accurately identifies the ground surface beneath the snow-cover. This advance encourages the collection of large multi-polarization GPR datasets for operational use by removing the subjectivity involved in the GPR post-processing and interpretation and alleviating the labour of manually interpreting radargrams through an objective function.

Sensitivity analysis showed how measurement errors propagate into the LiDAR – GPR measured snow density (Section 2.4.3). We found that measurement errors on the order of 10 cm for LiDAR and 1 ns for GPR may translate into errors in the density estimate of 150 kg/m$^3$ or greater, though these estimates were reduced to 30 kg/m$^3$ by median filtering and interpolating through outliers (Section 2.4.3). While the signal of each instrument is coherent, the leading source of error in our density measurement is spatial misalignments (potentially sourced from geolocation inaccuracies, point cloud to raster processing, and coordinate 440    transformations) that are on the scale of the 1 m resolution data products. In some locations the co-registration may be nearly exact between the two instruments, and the resulting error will be low. We found by cross-correlating the GPR and co-located



LiDAR snow depth transects, that misalignments of approximately 1 – 5 m are possible. To evaluate how spatial misalignments impact the training data, predictor data, and the regression model output, and to estimate the uncertainties introduced from integrating the cross-platform sensor data, we created multiple sets of training data by effectively perturbing where LiDAR—

GPR transects are aligned via cross-correlation lagging, and introduced common practice mistakes in the sensor integration, such as mixing the geographic coordinate system of the data between NAD83 and WGS84. We found that perturbing the sensor integration introduces less than 1 kg/m$^3$ error in the modelled density on average (up to 2 % in forest stands), that outlier filtering is robust to sensor integration errors, and this error is small relative to the overall SWE uncertainty.

Our work characterised the measurement uncertainties and the resulting SWE uncertainty in pursuit of the goal for 10 %

uncertainty in global SWE estimation (National Academies of Sciences, Engineering, and Medicine, 2018). Based on the evaluation of the remotely sensed or modelled snow properties with in situ measurements, we used simple linear regression to model uncertainties spanning both forested and open areas. The uncertainty in LiDAR snow depth varies spatially and is dependent on landscape characteristics such as slope and vegetation (Deems et al., 2013). However, our evaluation of snow depth in forested and open areas did not suggest that LiDAR snow depth errors were greater beneath the tree canopy (Table

1). The choice of uncertainties propagated through the SWE uncertainty analysis (Section 3.7) dictates which factor, depth or density, will have the greater contribution to the overall SWE uncertainty. Uncertainty in midwinter SWE tends to reduce at peak SWE, where snow depth and density are greater. Our findings (Fig. 10) are within the remarkably difficult to achieve 10 % goal and point to the success and accuracy of the joint LiDAR – GPR methodology for SWE retrieval at plot to forest stand scales.

We tested the model sensitivity to training and learned how much data is required for accurate density estimation. Using approximately 30,000 LiDAR – GPR derived densities (10 % of the total) from random subsets, we obtained density models that are statistically identical to those generated from the larger data set. Though random sampling is not a practical method for GPR data acquisition and analysis, this exercise showed that the amount of GPR information required to train the model parameters is not as important as collecting data in a variety of landscape and snow-cover characteristics. The large GPR grid

in open areas captured the high degree of spatial heterogeneity and improved LiDAR spatial predictor importance, while GPR acquired in forests added necessary data for estimating sub-canopy snow density.

The Grand Mesa IOP is one of the largest campaigns to examine the spatiotemporal patterns of SWE and provided a rich data source for snow density analysis. In most circumstances few snow pits are dug, and uncertainties arise from spatial sampling of the underlying density distribution. The distribution of density on Grand Mesa during the early February SnowEx 2020

campaign can be treated as a random normal variable with mean and standard deviation of $276 \pm 21$ kg/m$^3$, despite differences of roughly 25 kg/m$^3$ between the average density of forested and open areas. The large sample size of snow pits allowed us to accurately quantify the mean snow density for distributed SWE estimates. While the uncertainty in any measurement of density was found to be 2.5 % on average, we sought to quantify the degree of uncertainty in the SWE distributed from the sampled population of snow density as a function of sample size. Our analysis suggests that 10 spatially random snow pit observations

within the study domain are sufficient to reduce the median uncertainty in distributed SWE to within $10 \pm 2$ % (Appendix



B.2). Although the differences are marginal, we have shown that on average this simpler approach for distributing density more accurately represents the in situ observations than the SWE distributed using our modelled estimates of density (Section 3.6) but resolves no information about spatial patterns of snow density, and is therefore less useful for understanding density patterns across the landscape. In situ snow campaigns targeting average SWE require far fewer pits than needed to
resolve the spatial patterns. Without prior information regarding the correlation length of snow density, the random field synthesised density has unfounded spatial context, as the required covariance information cannot be retrieved by manual observations.

In situ snow density observations have limited spatial support and tend to examine shorter length-scales of variability than are expressed in distributed models or retrievable by radar remote sensing. As estimated from the variogram analysis, the median
distance between snow pit observations in our study is beyond the length scale of variability for snow density, and therefore the observations are spatially uncorrelated. Snow pits are an invaluable source of calibration and validation observations but do not adequately scale spatially, incur human errors and biases, and are time intensive to sample. For example, a team of two can fully sample a SnowEx pit in two hours, which for the approximately 100 snow pits in the study area, amounts to ~ 400 hours of labour (excluding the time to quality control (QC), curate the snow pit logs, and travel to and from the field site). The
160 km of GPR data used in this work required approximately 20 hours to collect and an additional 20 hours to QC TWTs, which amounts to ~ 40 hours, or roughly a 90 % reduction in field labour, excluding the labour for acquisition and processing of airborne LiDAR. Though it is worth noting the greater financial cost to obtain GPR equipment and outsource airborne LiDAR data collection. However, densities estimated from GPR TWTs and LiDAR snow depths are objective, repeatable, and offer the spatial continuity and areal coverage to provide insights to the spatial patterns of density.

The density measurements inferred from GPR profiles permitted us to quantify the spatial length scales of density variability, albeit insensitive to subpixel or sub-aggregated pixel length scales on which greater variability may exist. Using variogram analysis, we determined that measurements of density up to ~100 m are correlated. These findings significantly differ from a previous variogram analysis that found correlation lengths for snow density of less than 10 m (Yildiz et al., 2021). However, these findings are study site dependent, and it may be that we have identified an additional longer, lower spatial frequency
scaling of snow density. Our analysis of the correlation length of LiDAR snow depths generally agrees with scale-breaks identified in previous studies within forested and open areas (Deems et al., 2006; Marshall et al., 2006; Trujillo et al., 2009). Corollary to SWE, two-way travel-time in dry snow depends both on snow depth and density. We found that TWT and SWE consistently exhibited similar correlation lengths and nugget variability in the forested and open areas. This finding supports TWT as an informer of spatial SWE variability. We found that snow depth and TWT reached comparable maximum variability
in open areas, while depth variability is greater in the forested areas. Snow density exhibited 4 % greater variability in the open areas than in the forests, indicating that wind exposure increases the variability, and conversely, shelter provided by terrain and vegetation tends to reduce spatial density variability.

The LiDAR predictors were inspired by theory of wind-terrain-vegetation interactions governing snow distribution (Winstral et al., 2002). Though to keep the model design innate to LiDAR information, we did not include wind data or predictors such





as maximum upwind slope and wind factor. Instead, these wind parameters were utilised as a corroboratory metric for explaining spatial patterns predicted in regression modelling. For each regression model, we identified the most important LiDAR features used to distribute density and found dependencies on predictor importance due to model choice and architecture (Appendix A.2). Vegetation height and proximity to vegetation greater than 0.5 m in height appeared prominent in the three regression models, whereas dependence on elevation, slope, and aspect for snow density at Grand Mesa was weakened. We used a "kitchen-sink" approach to the regression modelling presented but found comparable accuracy in models using fewer parameters. To capture the range of processes (i.e., elevation, slope, aspect, and forest attributes) that influence snow densification, one field campaign in the western U.S. collected density measurements from 300 snow cores at $10-20$ m intervals and 17 snow pits (Broxton et al., 2019). From these observations, bulk snow density was distributed at 1 m resolution using an artificial neural network combined with airborne LiDAR-derived snow depth to estimate SWE. Broxton et al. (2019) highlighted the importance of representing the broader landscape with distributed densities for estimating SWE, finding ~30 % differences between the distributed estimates and observations from a nearby SNOTEL station. Elder et al. (1998) used a simpler, three feature (net radiation, slope, and elevation, with an intercept) MLR model that was trained on density observations of five snow pits and averages of five snow core transects to predict basin-wide average density and SWE. More recently, a similar study used a sampling strategy to represent unique classes of basin-wide physiography, acquiring ~1000 snow core observations, and used MLR and binary-classification tree models to distribute density from elevation and incoming radiation (Wetlaufer et al., 2016). The dependence of density on net solar radiation may explain the good performance of these models, whereas terrain parameters, such as slope and aspect, indirectly relate to radiation. The low relief and shallow slopes of Grand Mesa remove much of the elevation and aspect dependency on snow density but evince the interactions between wind, local terrain, and vegetation. Our approach in validating wind effects on density is an explanatory simplification of the controls on snow density which may be further impacted by forest stands. Effects such as the blocking of short-wave and emitted long-wave radiation from forest canopies, delivery of canopy intercepted snow to the snowpack, or the loss of snow mass and thereby altered compaction due to sublimation of canopy intercepted snow were unaccounted for (Bonner et al., 2022).

The regression models developed from the LiDAR and GPR acquisitions during the SnowEx 2020 Grand Mesa IOP will likely have weak predictive capability at other field sites. The SWE predictions, here, represent a single snapshot in time of snow depth and density. The patterns in depth, density, and SWE may be characteristic for mid-winter, dry snow conditions but other times of the year may exhibit different spatial patterns in all three (e.g., due to variable melt or liquid water during ripening). It may be necessary to recalibrate the model using GPR or another instrument to measure radar travel-time such as airborne FMCW radar (e.g., Yan et al., 2017). Provided radar-measured snow surface elevations, it is possible to infer density along FMCW flight-lines when combined with a snow-free digital surface model. However, high resolution elevation and snow depth data significantly improves modelled spatial heterogeneity in snow density. The expense of acquiring airborne remote sensing data is a crux of the technique, and it may not be feasible to fly entire catchments across the breadth of snow climates. Less expensive techniques for estimating SWE distribution, such as drone-based radar retrievals of dielectric





permittivity (e.g., Valence et al., 2022), and in situ measurement campaigns combined with learned-regression models (e.g., Wetlaufer et al., 2016; Broxton et al., 2019) should be utilised where appropriate and examined for the physical basis. Empirical models of this type are often distributed over vast areas with little validation or consideration to the underlying physical processes.

## 5. Conclusion

We developed an innovative approach to estimate SWE across a ~16 km$^2$ domain by evaluating GPR travel-times for bulk snow density given a snow depth constraint, then extrapolating across the domain using machine learning. Our automatic and objective technique for interpreting radargrams reduces post-processing labour, which is a primary hindrance to widespread use of GPR in snow science. We leveraged LiDAR estimated snow depth to solve for snow density along ~160 km of GPR transects. From these along-track estimates, we calculated the length-scales of variability for depth, density, SWE and found that snow depth had 25 % reduced variability, SWE had 12 % reduced variability, and density had 4 % increased variability in open areas compared to that exhibited within forested stands. In dry snow, we found TWT informs SWE, better than either depth or density independently. Snow density distributed by regression techniques revealed anomalies associated with localised terrain features and forest stands that shelter the snowpack from wind densification. Spatial patterns show the best agreement in the direction of prevailing winds strong enough to transport snow. Roughly 60 % of density variability in our single mid-winter survey can be accounted for using a wind factor analysis. Additionally, we estimated that snow density and depth contribute equally to SWE uncertainty for a typical mid-winter snowpack on Grand Mesa, Colorado, USA. On average, distributed relative SWE uncertainty was less than 10 % and tends to be greatest in the shallower and lower density snow beneath tree canopies. The total SWE is not observable, making it unclear which method is more accurate, yet analysis suggests that one snow pit per km$^2$ may reduce the SWE uncertainty to within $10 \pm 2$ %. However, using such a sampling strategy would not resolve spatial patterns and variability in snow properties. This pilot study provides a useful method resolving explanatory spatial patterns in snow depth, density, and water equivalent with comparable uncertainty to in situ methods but with spatial continuity at resolutions practical for calibration or validation of space-borne radar remote sensing retrievals of SWE.

## Appendix A.

### A.1 Regression Parameter Optimization

#### A.1.1 Multiple Linear Regression

The MLR model has the form

$$y = X\beta + \epsilon,$$ (A.1)



where $y$ is the observed density along the GPR transects, $X$ is a matrix with columns containing the normalised LiDAR predictors at the coordinates along the GPR transects, $\beta$ is the vector of the regression coefficients which we seek to estimate, and $\epsilon$ represents the model residual. From the method of least squares, the multiple linear regression coefficients are estimated as

$$\beta = (X^T X)^{-1} X^T y \ . \tag{A.2}$$

Using cross-validation to assess the model accuracy and sensitivity, we estimated the MLR model parameters. We trained the model with 1000 Monte Carlo simulations by randomly sampling 90 % of the density observations and testing on the remaining 10 %. Additionally, we repeated this process and randomly sampled only 10 % of the data and tested on the remaining 90 %. In doing so, we created two sets of parameters that robustly span the parameter space. Using these regression coefficients, Equation A.1 is computed to distribute the predicted densities. The modelled densities are insensitive to the training choice for parameter estimation, as the RMSD between the two models is less than 1 kg/m³.

### A.1.2 Random Forest and Artificial Neural Network Regression

Whereas MLR models are relatively inflexible and model overtraining is not a concern, techniques such as Random Forest and Artificial Neural Network regression are highly tuneable and may overfit the data. Hyperparameters determine the model architecture which is often designed subjectively or through an optimization process. The number of trees and the minimum leaf size of a tree were the hyperparameters adjusted for the Random Forest method. Neural networks offer a greater hyperparameter space, allowing for design of the number of and size of hidden layers, the neuron activation function, and model regularisation. The machine learning models were implemented using the MATLAB Regression Learner toolbox, where it was determined that model hyperparameters which minimise the cross-validation mean squared-error overfit the data. Model overfitting was remedied by training ensembles of models with various hyperparameters, calculating the averaged standard deviations of density data for each ensemble predicted along the GPR transects ($\overline{\sigma_{train}}$) and the densities predicted elsewhere in LiDAR domain ($\overline{\sigma_{pred}}$) and the coefficient of determination ($R^2$) of the training data and prediction. Optimal hyperparameters which do not overfit the data were then determined by minimising the objective function

$$\varphi = \frac{1}{R^2} \frac{\overline{\sigma_{pred}}}{\overline{\sigma_{train}}} \ . \tag{A.3}$$

The ratio of the standard deviations asserts that an appropriate model will have similar variance throughout the modelled domain, by penalising overfit data in the training locations, yet rewarding the model which explains the data accurately. We found that the best model parameterizations that are not overfit scored an $R^2 \cong 0.8$ with $RMSE \cong 15$ kg/m³. The corresponding hyperparameters for the Random Forest Regression were 10 trees with a minimum leaf size of 200. The ANN architecture had two hidden layers each with 50 neurons and hyperbolic tangent activations and regularisation of $\lambda = 0.015$.





### A.2 Predictor Importance

#### A.2.1 Multiple Linear Regression

We applied the "kitchen-sink" approach because the model that was trained using the LiDAR–GPR densities, which utilised every LiDAR predictor, exhibited the largest correlation ($R^2 = 0.27$) to the observations. However, various model parameterizations which utilised few parameters yielded equivalent accuracies. To assess the importance of the individual predictors, we assembled all combinations of 1 to 17 predictor models, solved the regression for each combination, and cross-validated against a test set of the LiDAR–GPR estimated densities. We considered optimal models as the top 1 % of outputs, and from these we tracked which predictors composed any model. We identified the relative importance of each predictor (Fig. A.1) by summing the number of appearances for a given predictor and dividing by the number of optimal models. Vegetation parameters and the east-west gradient of the ground surface elevation were featured in all the most accurate modelled predictions. Notably, snow density on Grand Mesa exhibits weak dependence on elevation, aspect, and slope.

#### A.2.2 Random Forest Regression

The permutation accuracy importance was calculated to determine which of the LiDAR derived predictor variables are most valuable in predicting the response. The permutation importance is assessed by comparing the accuracy of the prediction for a given learner (tree), then randomly permuting the predictor variable of interest and recalculating the prediction accuracy (Hapfelmeier et al., 2014). An important predictor will lose predictive capability after random permutation, while an unimportant predictor will be unaffected by the randomization. The prediction accuracy is calculated using "out of bag" observations that were excluded from the population used to build the decision tree (Breiman, 2001). The relative "out of bag" predictor importance for the ensemble of Random Forests generated using 10 trees with a minimum leaf size of 200 suggests that the slope of snow depth, snow surface elevation, ground surface elevation, proximity to vegetation, and vegetation height are the five leading predictors of snow density (Fig. A.2).

#### A.2.3 Artificial Neural Network Regression

Approaches which partition the weights between neural connections to determine the relative importance of the predictors within an ANN have a classically simple architecture with a network expressing one hidden layer (Goh, 1995). This technique becomes obfuscated when applying an ANN with multiple hidden layers.  To determine the relative importance of predictors within an ensemble of networks with two hidden layers we simply multiplied the matrices of weights connecting the input to the first hidden layer, the first to the second hidden layer, and second hidden layer to the output. The greater the overall weighting that is assigned to a predictor the greater the importance of the feature. This method suggests that vegetation height, proximity to vegetation, the north derivative of the ground surface elevations, the slope of the ground elevations, and east derivative of the snow surface elevations are the five leading predictors of snow density (Fig. A.3).





### A.3 Model similarity intercomparison

Visual inspection reveals apparent structural similarity among the three regression-based models. As a quantified model intercomparison we applied the coefficient of determination measured by Pearson correlation calculated on a pixel-by-pixel basis. The structural similarity index (SSIM; Wang et al., 2004) is a normalized value between 0 and 1 that is defined by image luminance, contrast, and standard deviation. We calculated the SSIM in 100 m radius kernels (comparable to the estimated correlation length of density) as a second means of determining the similarity among the model ensembles. Capturing various structural length scales examines the model similarities throughout the correlation length. Table A.1 overviews the $R^2$ similarity matrix. Nearly 50 % of the features observed in the MLR model are explained within the RF model, and vice-versa. The SSIM similarity matrix suggests greater structural similarity at larger spatial support (Table A.1). The random field synthesised model exhibits no repeatable structure and is uncorrelated from the various regression models, as expected since it was randomly generated from the overall snow density statistics only.

### Appendix B.

### B.1 Statistics of in situ, LiDAR – GPR Inferred, and Modelled Snow Density

To show that the inferred and modelled densities are within the range of measurements observed in the snow pits we provided the distribution of these three data sets for the entire study area domain (Fig. B.1). The means of the distributions are overlapping within the standard deviations of the datasets. The LiDAR – GPR inferred densities suggest a broader distribution of densities than observed in pits or modelled. Sampling biases may explain small disagreement between mean values. The sample size and spatial representation of each data set varies on many orders of magnitude. The distributions are unequally represented by vegetation class, as 18 % of the snow pits (17 of 96), 23 % of the modelled domain (3,665,343 of 15,753,500), and 7 % of the GPR transects (19,978 of 278,627) were located within the forest stands. However, other than a bias of - 10 kg/m$^3$ the distribution of modelled estimates closely resembles that of the snow pit measurements. Z-tests confirm that the snow density data are normally distributed about the mean and standard deviations listed with high confidence.

### B.2 Sample Uncertainty of Density and SWE

The Grand Mesa IOP is one of the largest campaigns to examine the spatiotemporal patterns of SWE and provided a rich data source for snow density analysis. In most circumstances, an order of magnitude fewer snow pits are available, and uncertainty arises from spatial sampling of the underlying density distribution. We found that snow pit density measurements have an average uncertainty of 2.5 %. To estimate the uncertainty in average density due to sample size and to propagate this uncertainty in terms of SWE, we conducted Monte Carlo simulations by randomly subsampling density observations. Utilising 1,000 Monte Carlo realisations as a function of sample size, we incrementally increased the number of randomly sampled snow pits to estimate the mean and the standard deviation of the distributed average snow density (Fig. B.2). We set the
sampled standard deviation as the spatial uncertainty in density and summed in quadrature the distributed errors in LiDAR snow depth as described in Section 3.7 to propagate the SWE uncertainty (Fig. B.2). Our analysis suggests that 10 snow pit observations are sufficient to reduce the median uncertainty in SWE to within $10 \pm 2$ %.

**Appendix C.**

**C.1 Ancillary Data: Wind Speed and Direction**

Figure C.1 presents the wind speed and direction data from the Mesa West meteorological station (Houser et al., 2022). Table C.1 lists the accompanying data presented in Fig. 8.

**Code availability**

The processing software developed for the multi-polarization GPR analysis is available at https://github.com/tatemeehan/SnowEx2020_BSU_pE_GPR.

**Data availability**

Snow pit observations (Vuyovich et al., 2021), snow depth observations (Hiemstra et al., 2020), and GPR travel-time observations (Meehan, 2021; Webb, 2021) acquired during the SnowEx 2020 Grand Mesa IOP are publicly available through

the National Snow and Ice Data Center (NSIDC). Data products resulting from this work, comprised of the LiDAR–GPR estimated density observations; LiDAR estimated snow depth, terrain, and vegetation model predictors, the ensemble of modelled snow density, and the SWE and uncertainty estimated therefrom will be archived within the NSIDC at DOI: 10.5067/LANQ53RTJ2DR pending the review of this manuscript.

**Author contribution**

HPM, CH, and KE organized and led the SnowEx 2020 Grand Mesa IOP; TM, AH, HPM, DM, RW, RB, CH, and KE collected field measurements and curated various datasets utilized in this work; HPM, ED, CH, and KE acquired funding for the field effort and labour in preparing this manuscript; TM, AH, HPM, RB, MR, and KE conceptualized the goals of this research; TM developed the methodology, conducted the formal analysis and prepared the manuscript; all co-authors provided feedback to the manuscript through review and editing.

**Competing interests**

The authors declare that they have no conflict of interest.



## Acknowledgements

This research was funded under the U.S. Army Engineer Research and Development Center (ERDC) Basic Research Program through Program Element 601102/Project AB2/Task 1. Permission was granted by the Director, Cold Regions Research and Engineering Laboratory, to publish this information with unlimited distribution. The findings of this document are not to be construed as an official Department of the Army position unless so designated by other authorized documents. The use of trade, product, or firm names in this document is for descriptive purposes only and does not imply endorsement by the U.S. Government. Funding was also provided by the NASA Terrestrial Hydrology Program (THP) in coordination with the SnowEx program awards NNX17AL61G and 80NSSC18K0877. We wish to thank all the scientific participants and logistical support staff of the SnowEx 2020 campaign, who helped make this research possible.

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



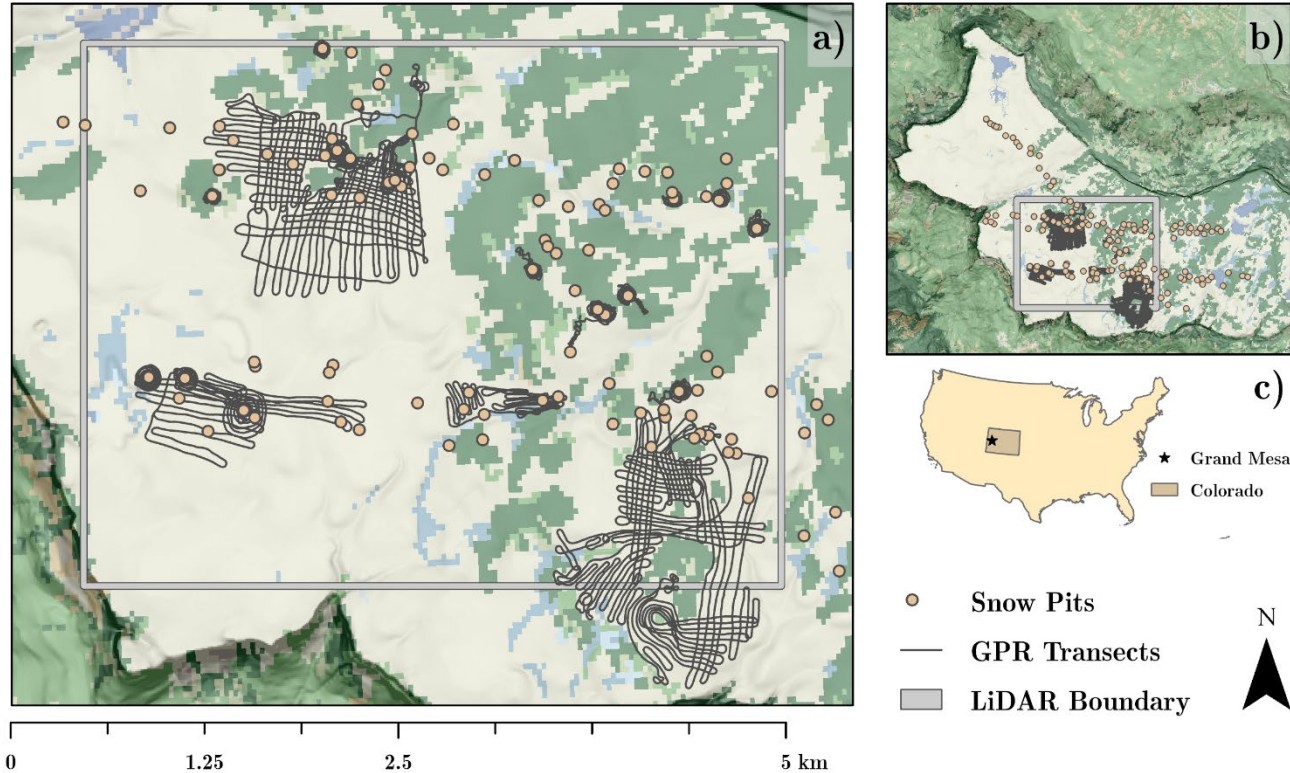

**Figure 1: a) Study area map of snow pit locations (yellow circles), GPR transects (black lines), and LiDAR boundary (grey wireframe). Landcover classification identifies the forested areas as green and lakes as blue. b) inset map of Grand Mesa, Colorado depicting the extent of the dataset acquired during the NASA SnowEx 2020 Intensive Observation Period at Grand Mesa, Colorado (Hiemstra et al., 2021). c) inset map of the contiguous U.S. which identifies the location of Grand Mesa, Colorado. Land cover classification data were accessed from the 2016 National Land Cover Database (Homer et al., 2020). Slope hillshade data were accessed from the USGS 3D Elevation Program (Lukas & Baez, 2021). Cartographic boundary files were accessed from the Census Bureau's MAF/TIGER geographic database (U.S. Census Bureau, 2020). The geographic coordinate projection of these maps is UTM Zone 12 N; EPSG code 32612.**





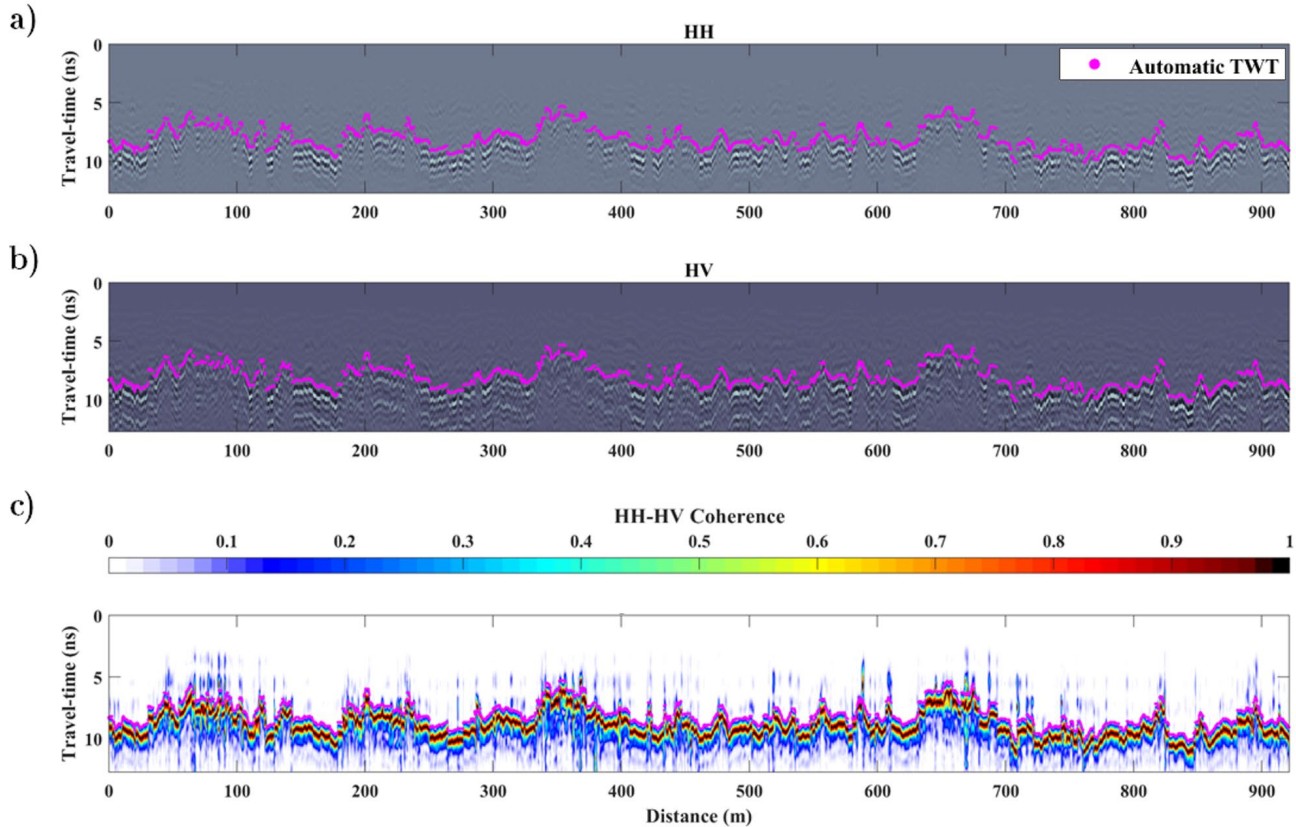

**Figure 2: A 900 m GPR transect with autopicks in magenta for a) HH and b) HV profiles of travel time, and c) the coherence of these radargrams (Equations 1, 2).**


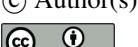
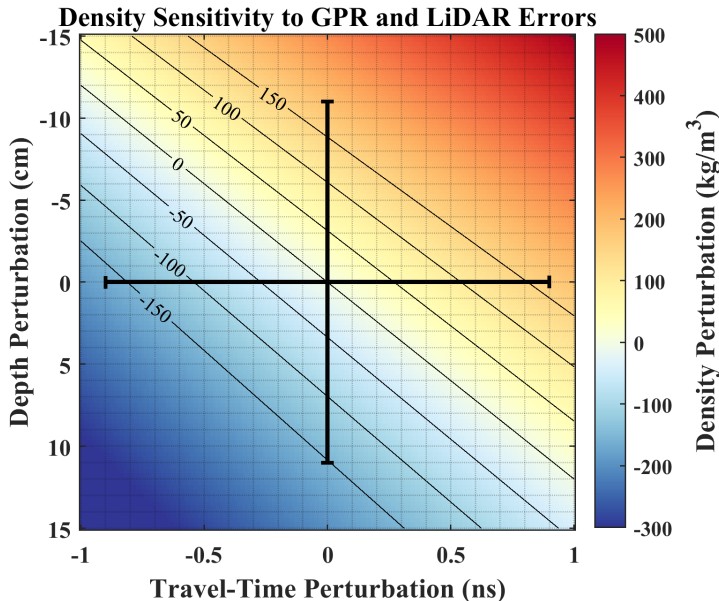

**Figure 3: Perturbations were added to the mean values 8 ns TWT and 96 cm depth then the density was evaluated to estimate potential errors resulting from sensor integration (coloured area and contours). The error bars represent the RMSE of LiDAR evaluated by probing and the RMSD of the GPR TWT crossovers. At one standard deviation, combined snow density errors of ± 150 kg/m³ can be expected from sensor integration, which are reduced to within ± 30 kg/m³ by outlier filtering.**

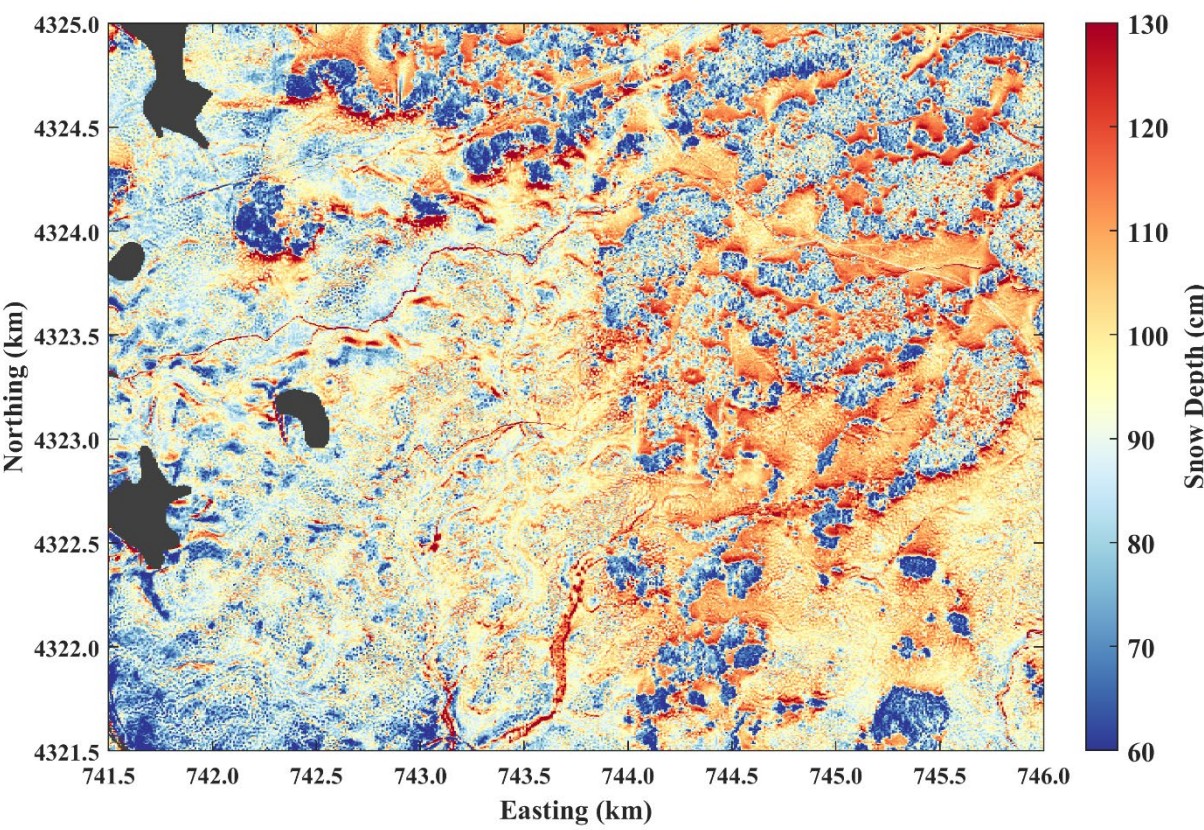

**Figure 4: One-metre resolution snow depths from the February 1, 2020 LiDAR flight. The western half of the domain is relatively unforested, shrub steppe (lakes are masked black) while the eastern half has stands of dense forest (see Fig. 1).**






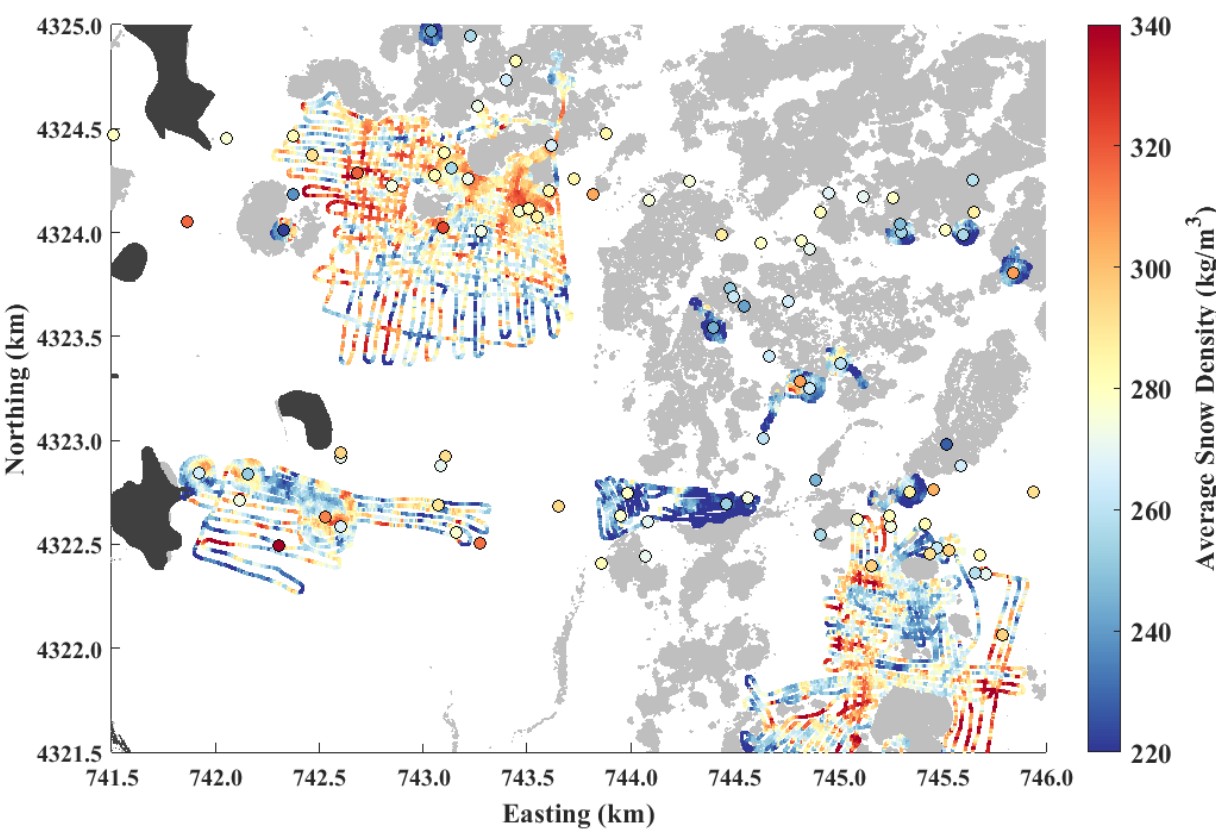

**Figure 5: Bulk snow density along radar profiles estimated by combining LiDAR snow depths with GPR TWTs. Average density measured in the 96 snow pits within the LiDAR boundary are overlaid as larger makers. Forested areas (grey) and lakes (black) are shown.**



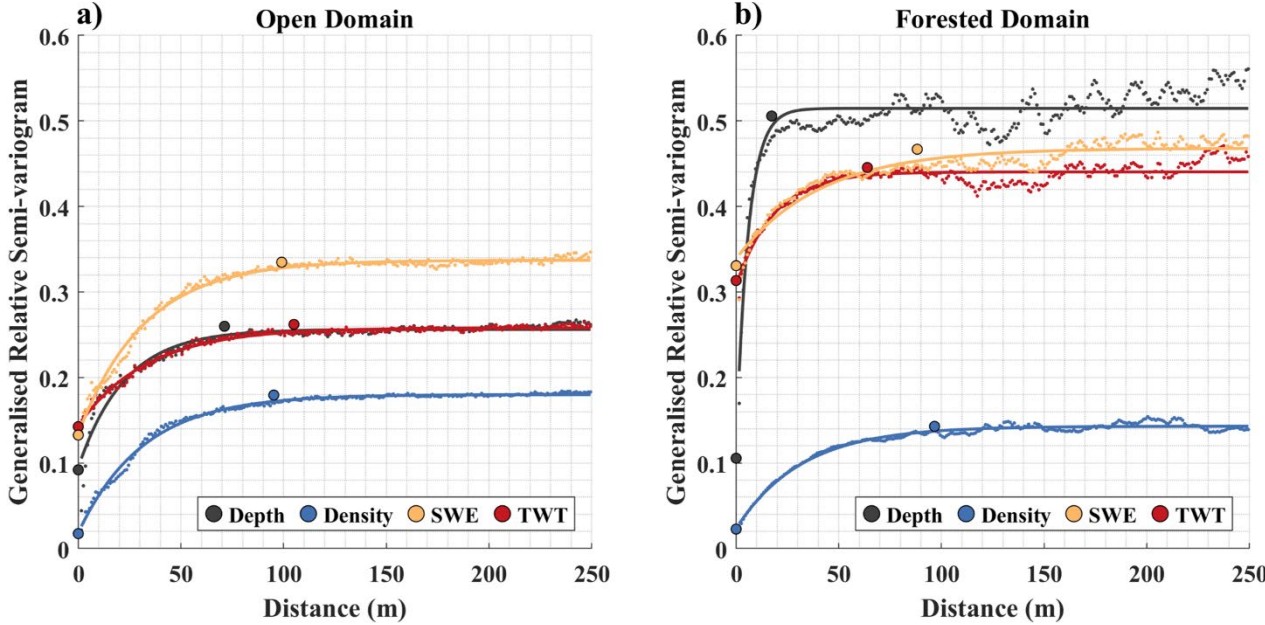


**Figure 6: Generalised relative semi-variograms in a) open and b) forested areas for LiDAR snow depth, GPR TWT, average density inferred along the GPR transects, and resulting SWE. Experimental variograms were fitted with an exponential model to determine the variogram parameters. The larger markers represent the average nugget, sill, and correlation length estimated by Monte Carlo subsampling.**

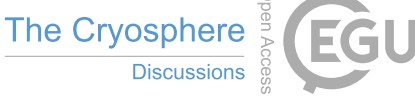



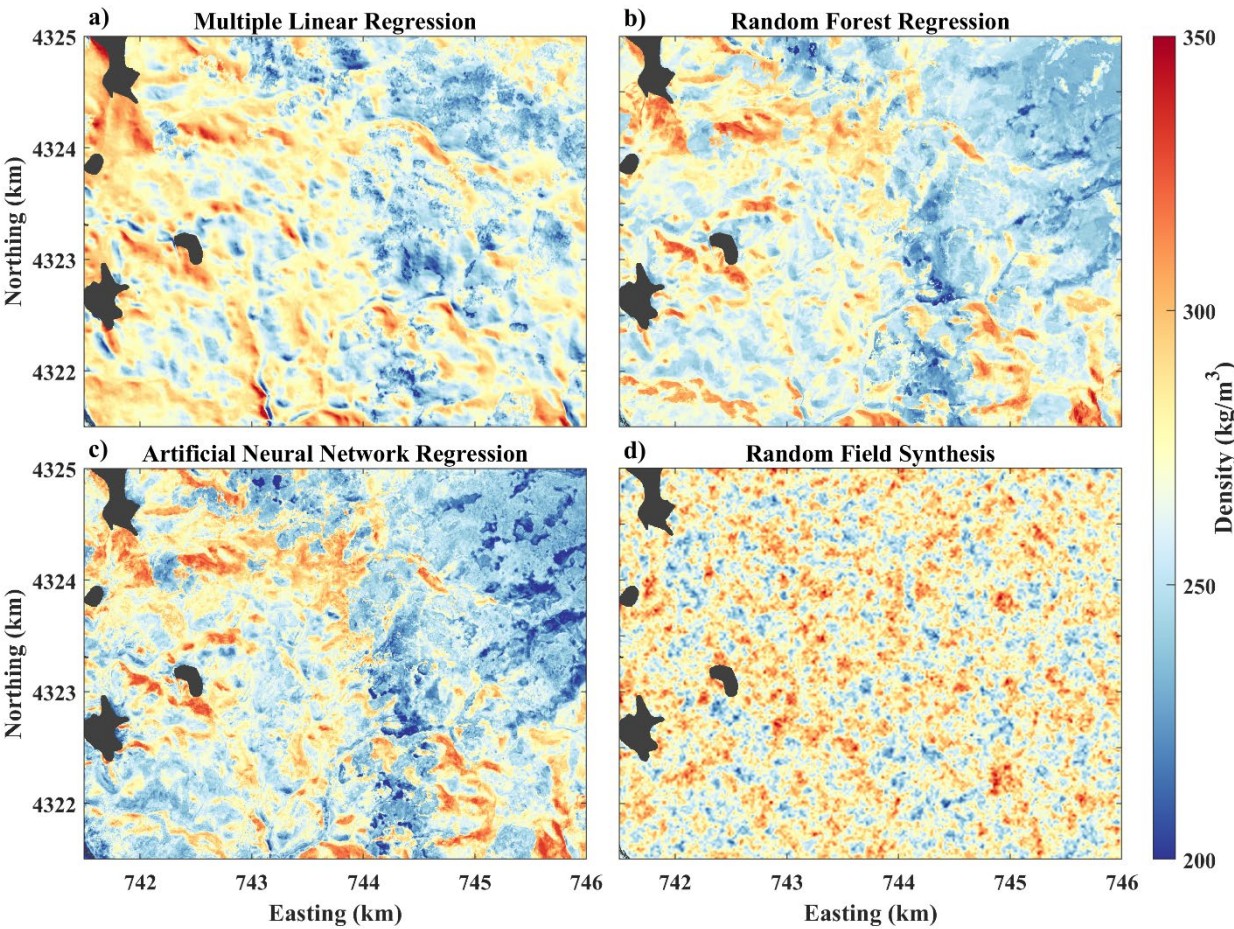

**Figure 7: Snow density was distributed spatially using regression techniques a) MLR, b) RF, and c) ANN based on predictive features that were derived solely from LiDAR information (i.e. snow depth, elevation, slope, aspect, and vegetation) and d) a synthesised random field model from the statistics of in situ density measurements (275 ± 20 kg/m³) and the correlation length of snow density estimated via variogram analysis.**




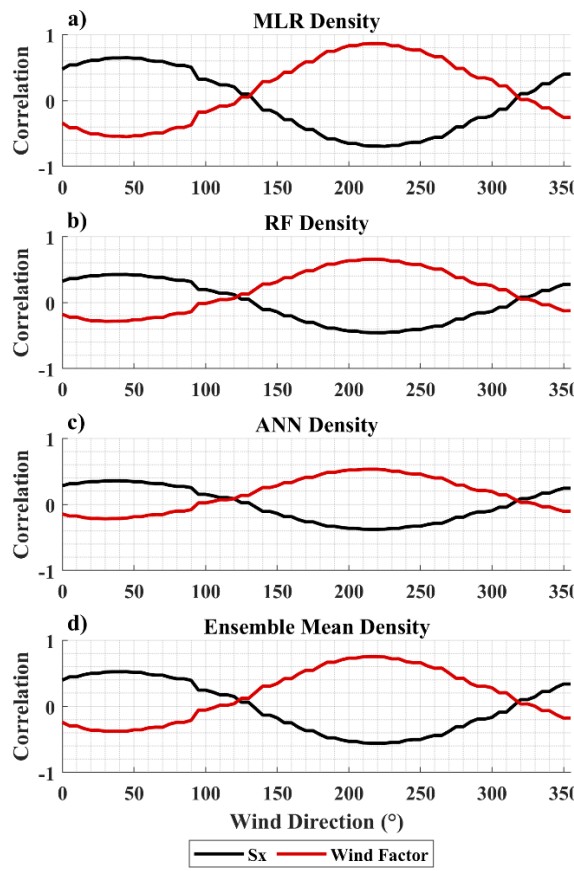


**Figure 8: Strong prevailing south-southwest winds (180° – 220°) provide a causal physical mechanism for correlations between the maximum upwind slope (Sx) and wind factor parameters calculated in 5-degree increments.**
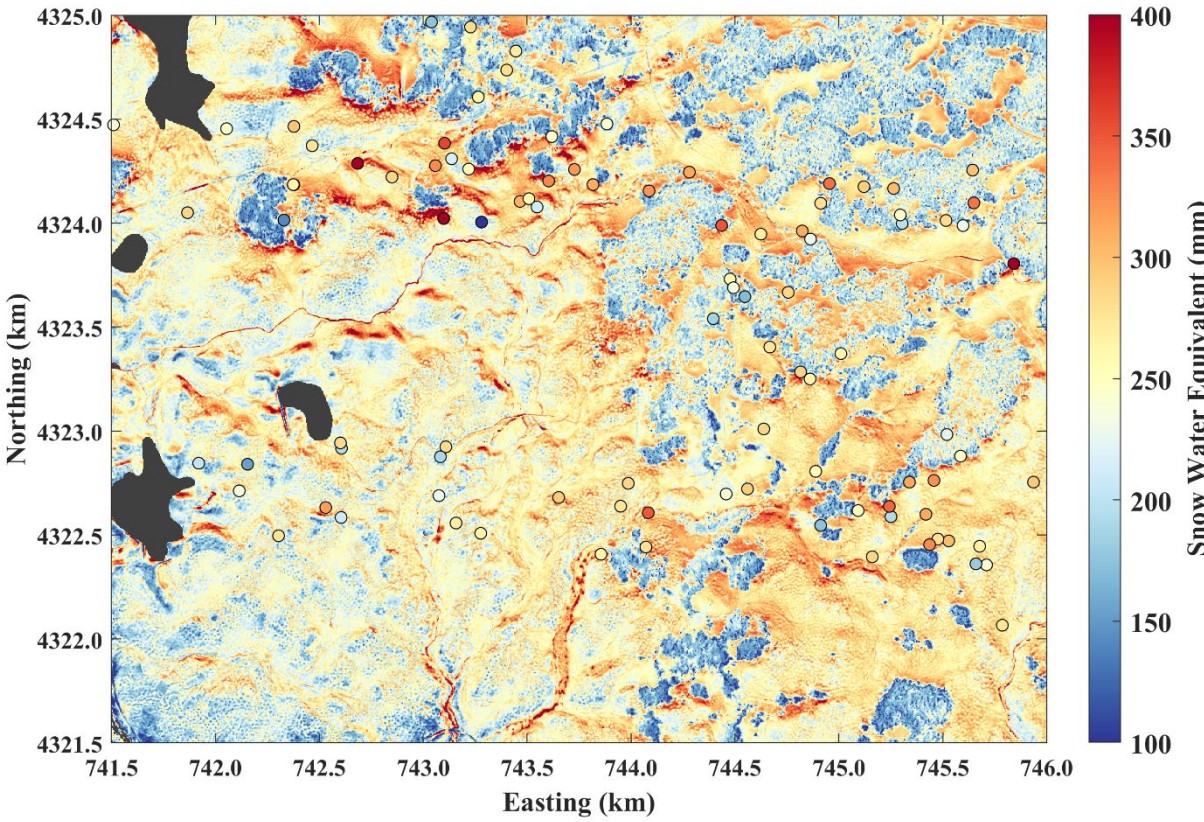

**Figure 9: Spatially distributed snow water equivalent estimated using the regression ensemble mean density and LiDAR snow depths. Forests and wind scoured areas tend to have lower SWE, and forest perimeters have higher SWE. The metre-scale stippled texture is the result of low-stature vegetation ($H_{veg} < 0.5$) and boulders, which both reduce snow depth and decrease the snow density. Large markers are SWE values measured at snow pits. Lakes are masked in black.**



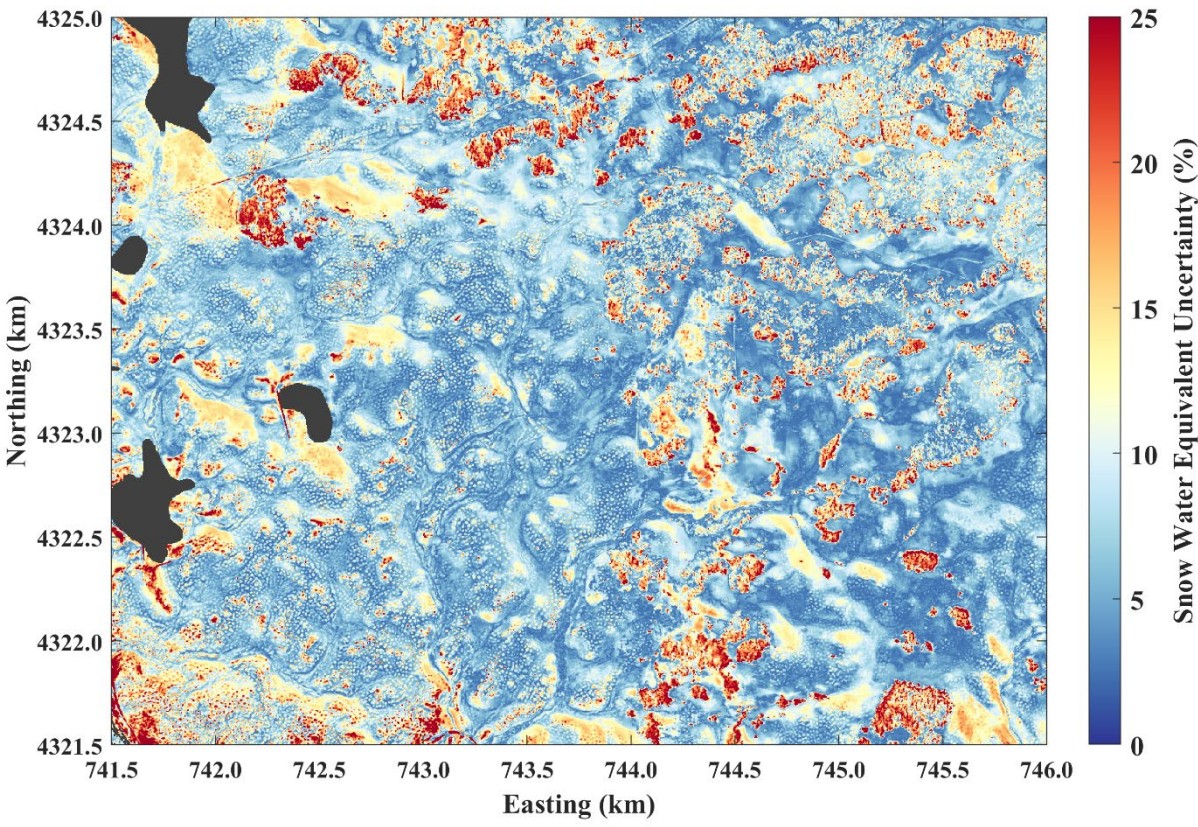

**Figure 10: Estimated uncertainty in snow water equivalent from the propagation of errors in snow depth and density. The spatial distribution of uncertainty tends to be greatest in the shallower and lower density snow underneath tree canopies and least in the deepest snow caught in drifts around the perimeters of forest stands.**

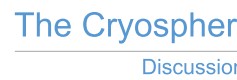
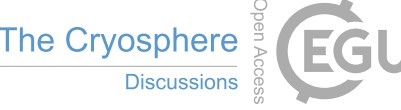

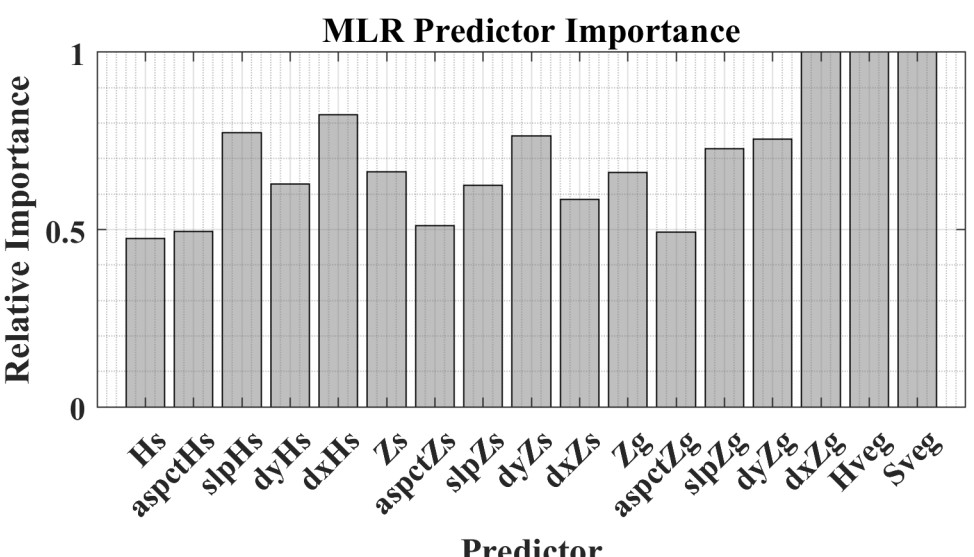

**Figure A.1: The relative importance of the LiDAR derived predictors: Hs snow depth, aspctHs aspect of snow depth, slpHs slope of snow depth, dyHs north component of snow depth gradient, dxHs east component of snow depth gradient, Zs the snow surface elevation and derivatives, Zg the ground elevation and derivatives, Hveg vegetation height, and Sveg the distance to vegetation with height greater than 0.5 m. The predictor importance was determined from the top 1 % of models.**


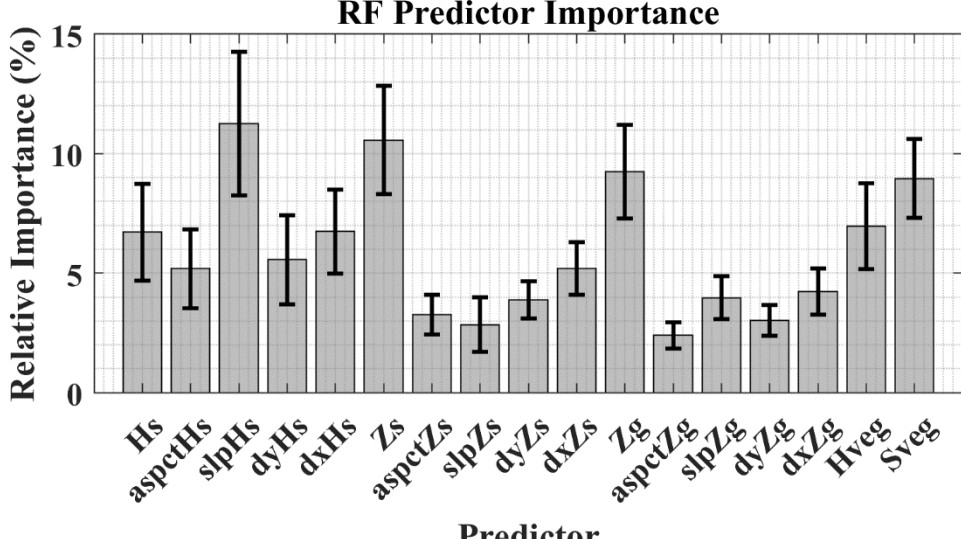

**Figure A.2: Relative importance of LiDAR predictors calculated using the "out of bag" technique for random forest regression. Uncertainties were developed using random subsets of training data in a Monte Carlo simulation.**






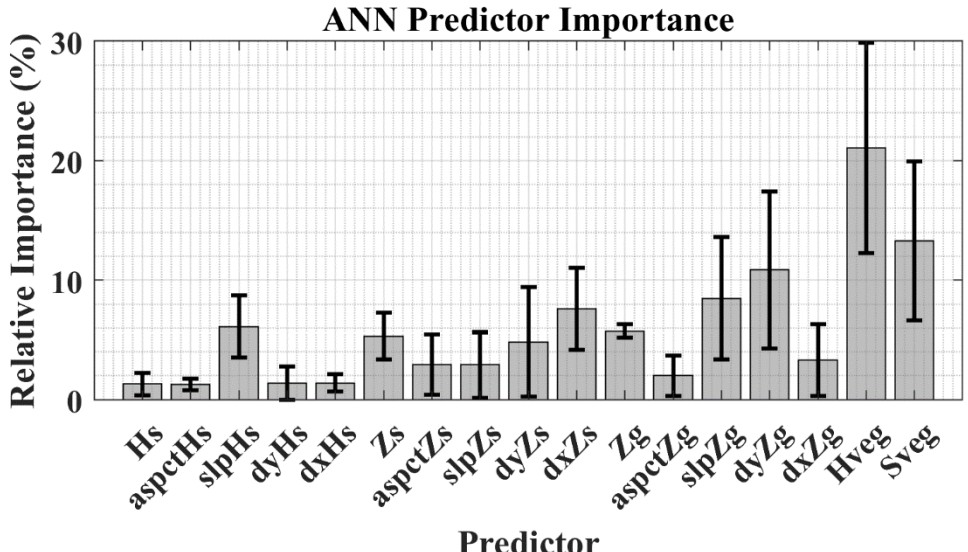

**Figure A.3: Relative importance of LiDAR predictors within the ANN comprised of two hidden layers.**

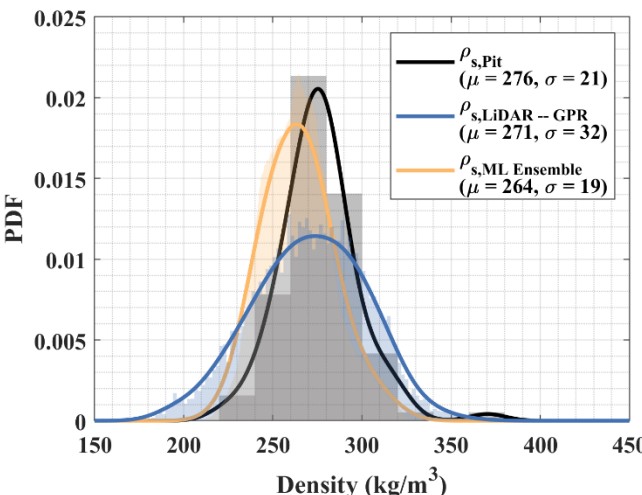

**Figure B.1: Histograms of the snow pit measured, LiDAR – GPR inferred, and regression model ensemble densities.**





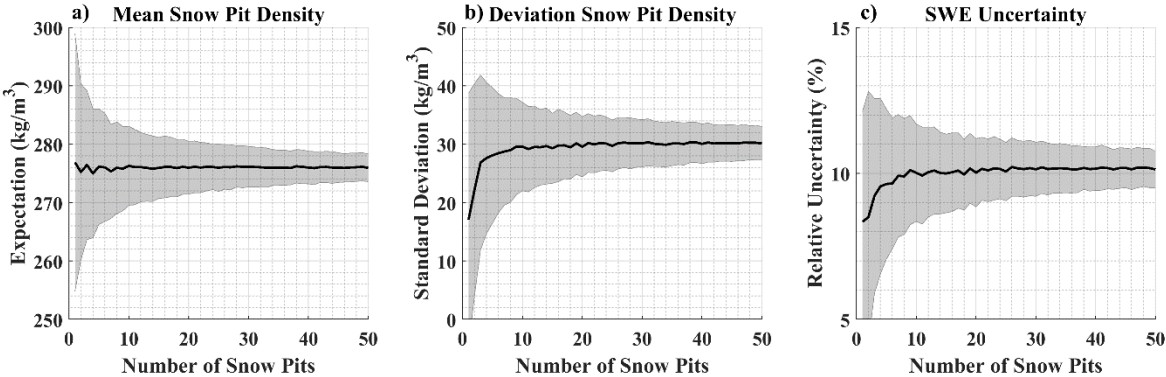

Figure B.2: Monte Carlo uncertainty analysis for a) mean snow density, b) standard deviation of snow density, and c) propagated SWE uncertainty as functions of sample size.

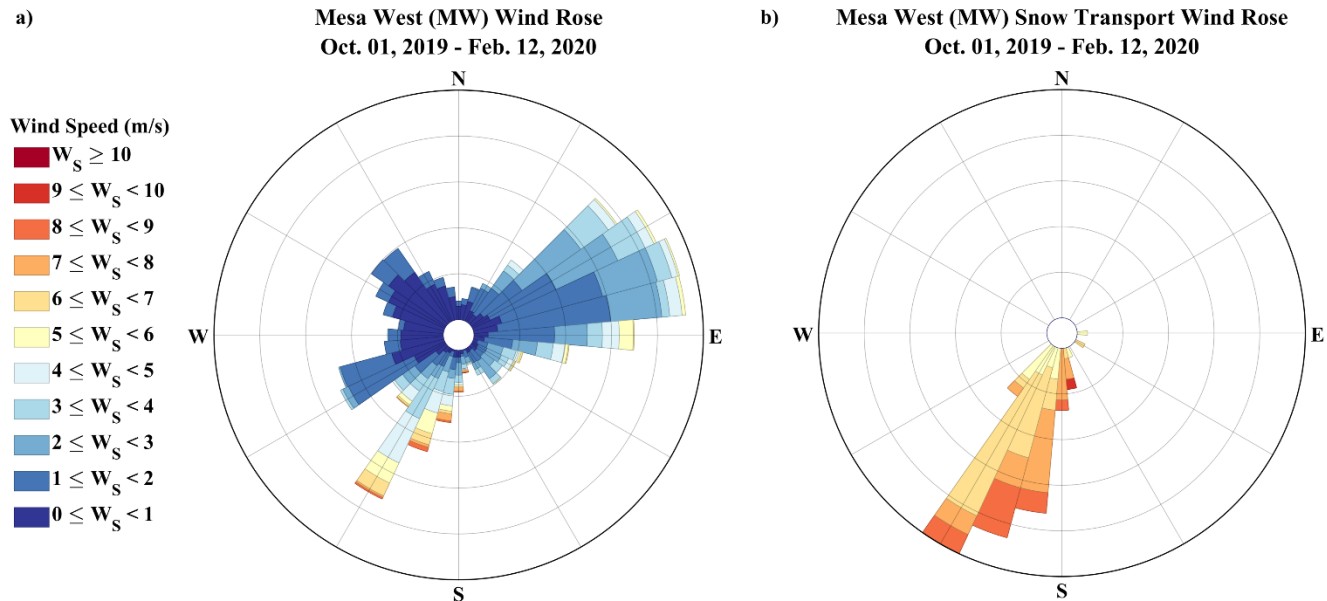

Figure C.1: Wind speed and direction data from the Mesa West meteorological station (Houser et al., 2022). a) wind rose spanning the beginning of the hydrological year through the end of the SnowEx IOP. b) wind rose of winds that are strong enough to transport snow (Li & Pomeroy, 1997) spanning the beginning of the hydrological year through the end of the SnowEx IOP. The median wind direction for snow transport is 200°.





**Table 1: a) The mean and standard deviation of snow pit and probe measured snow depths. b) Co-located estimates by LiDAR and GPR techniques gathered from within a 1 m radius of the in situ observations are compared to the in situ observations.**

**a)**

| Snow Depth (cm) Method | All Domain μ ± σ | Open Domain μ ± σ | Forested Domain μ ± σ |
|---|---|---|---|
| Snow Pit ($h_{s,Pit}$) | 97 ± 17 | 99 ± 17 | 87 ± 13 |
| Probe ($h_{s,Probe}$) | 95 ± 17 | 98 ± 16 | 84 ± 18 |

**b)**

| Snow Depth (cm) Method | All Domain μ ± σ \| $R^2$ \| RMSE \|Bias | Open Domain μ ± σ\| $R^2$ \| RMSE \| Bias | Forested Domain μ ± σ \| $R^2$ \| RMSE \| Bias |
|---|---|---|---|
| LiDAR ($H_{s,LiDAR}$) | 96 ± 16 \| 0.61 \| 11 \| 0 | 94 ± 16 \| 0.00 \| 22 \| -4 | 87 ± 14 \| 0.00 \| 23 \| 3 |
| GPR ($h_{s,\overline{Ens}\text{-}GPR}$) | 98 ± 17 \| 0.48 \| 18 \| 1 | 97 ± 14 \| 0.01 \| 21 \| -4 | 89 ± 12 \| 0.01 \| 24 \| 6 |

**Table 2: a) Mean and standard deviation for snow pit, LiDAR – GPR (Sections 2.4.1 and 2.4.3), and modelled densities. Snow pit**
**mean and standard deviations are estimated from all available data, N = 96 snow pits for the entire domain, N = 79 for the open domain, and N = 17 for the forested domain. b) Comparison between snow pit densities and estimated densities. Statistics ($R^2$, RMSE, and Bias) are measured from a subset of snow pits within 12.5 m of the GPR transects: N = 42 for all the domain, N = 36 for the open domain, and N = 6 for the forested domain. c) $\rho_{s,LiDAR-GPR}$ estimated densities are evaluated against modelled results.**

**a)**

| Snow Density (kg/m³) Method | All Domain μ ± σ | Open Domain μ ± σ | Forested Domain μ ± σ |
|---|---|---|---|
| Snow Pit ($\rho_{s,Pit}$) | 276 ± 21 | 280 ± 19 | 257 ± 20 |
| LiDAR – GPR ($\rho_{s,LiDAR-GPR}$) | 271 ± 32 | 273 ± 32 | 246 ± 19 |
| MLR Model ($\rho_{s,MLR}$) | 268 ± 21 | 273 ± 19 | 253 ± 18 |
| RF Model ($\rho_{s,RF}$) | 269 ± 17 | 271 ± 18 | 262 ± 11 |
| ANN Model ($\rho_{s,ANN}$) | 260 ± 25 | 266 ± 25 | 242 ± 14 |
| ML Ensemble ($\rho_{s,\overline{Ens}}$) | 264 ± 19 | 269 ± 19 | 248 ± 11 |
| Random Field ($\rho_{s,Rand}$) | 275 ± 20 | 275 ± 20 | 274 ± 20 |


**b)**

| Snow Density ($\rho_{s,Pit}$) | All Domain $R^2$ \| RMSE \|Bias | Open Domain $R^2$ \| RMSE \| Bias | Forested Domain $R^2$ \| RMSE \| Bias |
|---|---|---|---|
| LiDAR – GPR ($\rho_{s,LiDAR-GPR}$) | 0.02 \| 35 \| -5 | 0.01 \| 37 \| -8 | 0.39 \| 23 \| -13 |
| MLR Model ($\rho_{s,MLR}$) | 0.05 \| 26 \| -9 | 0.01 \| 26 \| -10 | 0.0 \| 26 \| -2 |
| RF Model ($\rho_{s,RF}$) | 0.01 \| 26 \| -8 | 0.0 \| 27 \| -10 | 0.0 \| 22 \| 4 |
| ANN Model ($\rho_{s,ANN}$) | 0.06 \| 30 \| -11 | 0.03 \| 31 \| -12 | 0.0 \| 23 \| -8 |
| ML Ensemble ($\rho_{s,\overline{Ens}}$) | 0.05 \| 27 \|-11 | 0.02 \| 28 \| -12 | 0.0 \| 22 \| -4 |
| Random Field ($\rho_{s,Rand}$) | 0.01 \| 27 \| -2 | 0.03 \| 25 \| -6 | 0.07 \| 32 \| 20 |





c)

| Snow Density ($\rho_{s,LiDAR-GPR}$) | All Domain R² \| RMSE \| Bias | Open Domain R² \| RMSE \| Bias | Forested Domain R² \| RMSE \| Bias |
|---|---|---|---|
| MLR Model ($\rho_{s,MLR}$) | 0.27 \| 27 \| 0 | 0.25 \| 27 \| 0 | 0.04 \| 22 \| 5 |
| RF Model ($\rho_{s,RF}$) | 0.80 \| 15 \| 0 | 0.79 \| 15 \| 0 | 0.76 \| 10 \| 1 |
| ANN Model ($\rho_{s,ANN}$) | 0.79 \| 15 \| 0 | 0.78 \| 15 \| 0 | 0.69 \| 11 \| 0 |
| ML Ensemble ($\rho_{s,\overline{Ens}}$) | 0.72 \| 18 \| 0 | 0.71 \| 18 \| 0 | 0.59 \| 13 \| 2 |
| Random Field ($\rho_{s,Rand}$) | 0.0  \| 32 \| 4 | 0.0  \| 37 \| 3 | 0.04 \| 39 \| 27 |


**Table 3: Variogram parameters are overviewed for the forested and open areas of Grand Mesa. a) The nugget indicates the percentage amount of measurement variability relative to the mean between co-located observations. b) The sill indicates the percentage variability relative to the mean at which measurements are no longer correlated (coefficient of variation). c) The correlation length is the distance in metres above which measurements are no longer correlated.**


a)

| Variogram Parameter: Nugget (% of mean) | Open Domain μ ± σ | Forested Domain μ ± σ |
|---|---|---|
| Depth ($h_{s,LiDAR}$) | 9 ± 0 % | 11 ± 2 % |
| Density ($\rho_{s,LiDAR-GPR}$) | 2 ± 0 % | 2 ± 0 % |
| SWE ($b_{s,LiDAR-GPR}$) | 13 ± 0 % | 33 ± 1 % |
| TWT ($\tau_{s,GPR}$) | 14 ± 0 % | 31 ± 1 % |

b)

| Variogram Parameter: Sill (% of mean) | Open Domain μ ± σ | Forested Domain μ ± σ |
|---|---|---|
| Depth ($h_{s,LiDAR}$) | 26 ± 0 % | 51 ± 2 % |
| Density ($\rho_{s,LiDAR-GPR}$) | 18 ± 0 % | 14 ± 0 % |
| SWE ($b_{s,LiDAR-GPR}$) | 33 ± 1 % | 47 ± 1 % |
| TWT ($\tau_{s,GPR}$) | 26 ± 0 % | 45 ± 0 % |

c)

| Variogram Parameter: Correlation Length (m) | Open Domain μ ± σ | Forested Domain μ ± σ |
|---|---|---|
| Depth ($h_{s,LiDAR}$) | 71 ± 4 | 17 ± 2 |
| Density ($\rho_{s,LiDAR-GPR}$) | 95 ± 2 | 97 ± 4 |
| SWE ($b_{s,LiDAR-GPR}$) | 99 ± 5 | 88 ± 17 |
| TWT ($\tau_{s,GPR}$) | 105 ± 6 | 64 ± 5 |

**Table 4: a) Mean and standard deviation of snow water equivalent measured in snow pits and distributed by LiDAR snow depths using an average snow pit density value (276 kg/m³), the average of snow pit density in each respective domain (276, 280, 257 kg/m³), the regression ensemble densities, and the random field densities. In situ mean and standard deviations are estimated from all available data, N = 96 snow pits for the entire domain, N = 79 for the open domain, and N = 17 for the forested domain. b) Evaluation**






of SWE between snow pit observations and distributed estimates using LiDAR snow depths multiplied by the average density measured from snow pits in the respective domains, the learned-regression ensemble modelled densities, and the random field synthesized densities.

**a)**

| Snow Water Equivalent (mm) Method | All Domain $\mu \pm \sigma$ | Open Domain $\mu \pm \sigma$ | Forested Domain $\mu \pm \sigma$ |
|---|---|---|---|
| Snow Pit ($b_{s,Pit}$) | 269 ± 57 | 278 ± 55 | 225 ± 45 |
| Average Pit Density (276 kg/m³) | 255 ± 51 | 266 ± 41 | 219 ± 62 |
| Domain Pit Avg. (276, 280 & 257) | 255 ± 51 | 275 ± 37 | 197 ± 44 |
| ML Ensemble ($b_{s,LiDAR-\overline{Ens}}$) | 245 ± 53 | 259 ± 41 | 198 ± 57 |
| Random Field ($b_{s,LiDAR-Rand}$) | 254 ± 54 | 265 ± 45 | 218 ± 64 |


**b)**

| Snow Water Equivalent (mm) Method | All Domain $R^2$ \| RMSE \|Bias | Open Domain $R^2$ \| RMSE \| Bias | Forested Domain $R^2$ \| RMSE \| Bias |
|---|---|---|---|
| Pit Average (276, 280 & 257) | 0.57 \| 38 \| -9 | 0.63 \| 34 \| -3 | 0.16 \| 55 \| -29 |
| ML Ensemble ($b_{s,LiDAR-\overline{Ens}}$) | 0.56 \| 42 \| -18 | 0.61 \| 38 \| -15 | 0.19 \| 56 \| -32 |
| Random Field ($b_{s,LiDAR-Rand}$) | 0.49 \| 42 \| -10 | 0.52 \| 39 \| -10 | 0.10 \| 55 \| -14 |

Table A.1: Similarity matrix of R² values for a pixel-by-pixel intercomparison and SSIM values for a model intercomparison estimated over 100 m radius (approximate correlation length of snow density).

| $R^2$, SSIM | MLR | RF | ANN | Random Field |
|---|---|---|---|---|
| **MLR** | 1, 1 | - | - | - |
| **RF** | 0.49, 0.61 | 1, 1 | - | - |
| **ANN** | 0.31, 0.46 | 0.39, 0.52 | 1, 1 | - |
| **Random Field** | 0, 0 | 0, 0 | 0, 0 | 1, 1 |


Table C.1: The maximum correlation and wind direction between snow density and the maximum upwind slope (Sx) and wind factor parameters. Data corresponds to Fig. 8.

| Snow Density Method | Sx | | Wind Factor | |
|---|---|---|---|---|
| | R | Direction | R | Direction |
| LiDAR – GPR | -0.45 | 225 | 0.48 | 220 |
| MLR | -0.69 | 225 | 0.86 | 220 |
| RF | -0.46 | 225 | 0.65 | 215 |
| ANN | -0.38 | 220 | 0.53 | 220 |
| ML Ensemble | -0.56 | 225 | 0.75 | 220 |
| Random Field | -0.03 | 180 | 0.02 | 170 |