# Peer review of "Spatially distributed snow depth, bulk density, and snow water equivalent from ground-based and airborne sensor integration at Grand Mesa, Colorado, USA"

_The Cryosphere, 2023_

## Referee Comment (RC2)

**Review of Spatially distributed snow depth, bulk density, and snow water equivalent from ground-based and airborne sensor integration at Grand Mesa, Colorado, USA**
**Meehan et al., The Cryosphere  Discussion**
César Deschamps-Berger

This article investigate various ways to spatialize snowpack bulk density based on density retrieved from ground penetrating radar (GPR) transect and quasi simultaneous snow depth retrieved from airborne lidar. The approach used is based on advanced technologies as opposed to labor-intensive, human made, snowpit measurements. The topic is very important in snow water equivalent estimation as more snow depth maps become available but the density remains rarely measured and even more rarely spatialized.

The work is interesting, well presented and will be of interest for people engaged in water resources estimations. I have the following concerns that should be adressed.

1. I am generally unsure about the way some statistics metrics and analysis are presented. A large importance is given to correlation, covariance, R and $R^2$ values in a too simplistic way (e.g. see comments about L340, L349). It is not clear what makes a $R^2$ value a proof of a significative correlation (see comments about L414). In addition, it is not clear if some correlation between snow densities and wind proxy are not expected considering the topographic predictors used to model the density (see comment about L353). Some additional figures would help to get a better feeling of the results, such as scatter plots between the estimated density, SWE, snow depth and the observed values at the pits.

2. The article is rich, maybe too rich, in methods and results which are diseminated all along the manuscript. For example results about the error estimation based on sensitivity analysis are given in the method (L247), a description of how other uncertainties are calculated is given in the results (L416-417), new methods are presented in the results (L346, L383) or discussion (L443). I would encourage the authors to move the different paragraphs in homogeneous sections.

**Minor comments**
L13 A bit surprising to focus here on spaceborne measurement while they are absent of the rest of the article.

L13 "*enables landscape-scale snow covered area*" can "*enable*" be grammatically used like that? I acknowledge that the coauthor team has a much better command of English than I do. I still pointed out some gramatical things that seemed odd to me. Please consider it as genuine questions.

L18 "*essential snow physics*" is a word missing?

L23 "*The root-mean-square error between the distributed estimates*" A doubt remains whether this RMSE is i) between modeled density and snowpit density measurement or ii)

the variability of the various models. In the latter case, I would not call it RMSE as there is no independant reference.

L24 maybe keep depth and SWE with the same unit, cm?

L25 "*Wind, terrain, and vegetation interactions display corroborated controls on bulk density that show model and observation agreement.*" I thought that one conclusion of the article was that snowpit are too sparse to sample the terrain variability.

L29 "*declining **of***"?

L35 "*from ground observations is not possible*" this should be tempered. See for instance

Molotch et al. 2004, "Estimating the spatial distribution of snow water equivalent in an alpine basin using binary regression tree models: the impact of digital elevation data and independent variable selection" https://onlinelibrary.wiley.com/doi/10.1002/hyp.5586

L38 "*instruments, which*" delete the ","?

L40 "WorldView" Snow depth maps were also calculated from Pléiades images (Marti et al., 2016; Shaw et al., 2019; Shaw et al., 2020; Deschamps-Berger et al., 2020; Eberhard et al., 2021). Replace WorldView with "high-resolution satellite stereo images" and maybe keep Marti et al., 2016 and McGrath et al., 2019.

Marti, R., Gascoin, S., Berthier, E., de Pinel, M., Houet, T., and Laffly, D.: Mapping snow depth in open alpine terrain from stereo satellite imagery, The Cryosphere, 10, 1361–1380, https://doi.org/10.5194/tc-10-1361-2016, 2016.

Shaw, T. E., Gascoin, S., Mendoza, P. A., Pellicciotti, F., and McPhee, J.: Snow depth patterns in a high mountain Andean catchment from satellite optical tristereoscopic remote sensing, Water Resour. Res., 56, e2019WR024880, https://doi.org/10.1029/2019WR024880, 2019.

Shaw, T., Caro, A., Mendoza, P., Ayala, Á., Gascoin, S., and McPhee, J.: The Utility of Optical Satellite Winter Snow Depths for Initializing a Glacio-Hydrological Model of a High-Elevation, Andean Catchment, Water Resour. Res., 56, e2020WR027188, https://doi.org/10.1029/2020WR027188, 2020.

Deschamps-Berger, C., Gascoin, S., Berthier, E., Deems, J., Gutmann, E., Dehecq, A., Shean, D., and Dumont, M.: Snow depth mapping from stereo satellite imagery in mountainous terrain: evaluation using airborne laser-scanning data, The Cryosphere, 14, 2925–2940, https://doi.org/10.5194/tc-14-2925-2020, 2020

Eberhard, L. A., Sirguey, P., Miller, A., Marty, M., Schindler, K., Stoffel, A., and Bühler, Y.: Intercomparison of photogrammetric platforms for spatially continuous snow depth mapping, The Cryosphere, 15, 69–94, https://doi.org/10.5194/tc-15-69-2021, 2021.

L42 LiDAR and photogrammetry

L43 "*whereas*" I do not see opposition between the two parts of the sentence.

L51 "*in space more significantly*" => "more in space"

L54: give the spatial variability scale of snow depth as well.

L65 Similar, other works looked at converting snow depth to SWE, see for instance and the citations therein:

Winkler, M., Schellander, H., and Gruber, S.: Snow water equivalents exclusively from snow depths and their temporal changes: the Δsnow model, Hydrol. Earth Syst. Sci., 25, 1165–1187, https://doi.org/10.5194/hess-25-1165-2021, 2021.

Fontrodona-Bach, A., Schaefli, B., Woods, R., Teuling, A. J., and Larsen, J. R.: NH-SWE: Northern Hemisphere Snow Water Equivalent dataset based on in situ snow depth time series, Earth Syst. Sci. Data, 15, 2577–2599, https://doi.org/10.5194/essd-15-2577-2023, 2023.

L71 "*more simple*" => "simpler"?

L116 "*from mean density measured in snow pits with the airborne LiDAR snow depths*"? not clear, please rephrase.

L131 Cite some of these works.

L202 Please repeat here the IOP date.

L212 What is NV5 Geospatial?

L210 "*the local distance between two point clouds*" This sounds like a significative difference with most snow depth studies (e.g. ASO Painter et al., 2016) in which the snow depth is calculated along the vertical direction as the difference between gridded products. I think it should introduce a discrepancy as well with the snow depth measured in snowpits.

L219 "*lidar*" written LiDAR elsewhere

L219 Could the statistics of the elevation difference over stable terrain (snow free, unchanged) be used to estimate the uncertainty?

L224 "*We applied a k-d tree searcher (Bentley, 1975) to co-register the LiDAR coordinates within a 1 m radius of the GPR TWTs*" I understand co-registration as moving one the elevation dataset for instance with a translation vector (X,Y,Z). Is it the case here? If so, can you cite an article using this k-d tree searcher algorithm? It does not seem very common.

L226 "*By calculating the maximum cross-correlation lag on continuous segments of transect data*" not clear what this operation is.

L239 2.4.3 Error and uncertainty are described at various places of the article (L247, L416-417, L443). This is confusing. Please reorganize it.

L249 "*are spatially uncorrelated suggesting that the errors are random and can be treated with filtering.*" = >"are spatially uncorrelated and can be treated with filtering." "filtering" is quite vague, not any filter would be adapted.

L252 "*we reduced the random error to ± 30 kg/m3 by filtering outliers.*" Not clear whether you optimized filtering until this error level was reached or if it is a fortunate consequence.

L256 When was the last snowfall prior the IOP?

L312-318 This paragraph does not match the title of this section. It could be move in 2.6, in a section gathering all the spatialisation methods used.

L327 What is the spatial resolution of the density maps? All this smoothing must reduce the actual resolution of the maps.

L340 "*The in situ snow depth observations (hs,Pit and hs,Probe) compare well with the LiDAR snow depths throughout the entire domain (R2 = 0.61, RMSE = 11 cm, ME = 0 cm), however within the open and forested domains individually, LiDAR and GPR estimated snow depths are uncorrelated with in situ snow depths (Table 1).*" This is puzzling. Can a set of points be correlated while two subsets including all the points are not correlated? A simple scatter plot, even in supplement, would help get a better feeling of the data agreement.

L346 "*The GPR data were acquired within a few metres of, but not directly beside the faces of snow pits, which necessitated a radius for pairing observations. The choice of 12.5 m matches that of the filtering during the sensor integration step (Section 2.4.3)*" This should be in method, not results.

L348 "*Measurements accumulated over 12.5 m distance introduce inherent variability on the order of 10 % (Section 3.3), which, along with differences between representative observation scales may explain the weak correlation between estimated density and the in situ measurements.*" This should go in discussion.

L349 "*weak correlation*" sounds contradictory with the 13% RMSE mentionned which sounds good.

L353 (and L383) Isn't there strong correlations between Sx, the Wind Factor and the topographic predictors used to estimate the densities? In that case, isn't it expected, by construction, that the densities and the wind proxies will be correlated?

L355 "**suggests** *that the method retrieves density patterns which are related to the degree of exposure and shelter due to topography and vegetation*" => to discussion

L361 "*The larger (roughly 10 m) spatial support of the LiDAR – GPR estimated densities cannot directly sense subpixel correlation lengths and **potentially** missed a zero-to-five-metre scale-break that is more comparable to the spatial support of in situ density observations*" The existence of the scale break sounds like speculation/discussion, not results. Besides, it sounds obvious that a measurement with a support of N m cannot be used to measure spatial variability at scale < N m.

L364 "*the expected variability among co-located $\rho$ is approximately 2 %,*" Based on which semi-variogram? Not clear. Again, this sounds more like discussion to me.

L367 "*to resolve spatial patterns*" => "to resolve spatial patterns at finner scale." There might be spatial pattern of larger scale than 100 m.

L368 "*depth and density **formulate** TWT and SWE,*" formulate is not clear. By definition, the SWE is the product of depth and density.

L370 3.4. Please provide metrics comparing the density modeled and measured (mean, median bias, standard deviation, RMSE...).

L376 "*This coincided*" what is "this"?

L378 "*as the random field contains little meaningful spatial information*" This is quite expected (idem L388). By construction it is spatially random. I am not sure what is the point of this randomly distributed density. Is it used in other studies?

L379 "*spreads the strengths*" formulation unclear.

L391 "*The lack of a large-scale topographic trend in density, such as one driven by elevation or aspect, evinces the role of forest vegetation on density.*" is this supported by a figure showing density against elevation or aspect? I don't understand the link between the two parts of the sentence, the elevation/aspect and the forest vegetation.

L407 "*with an appropriate correlation length and prior mean and spread but maintain a larger bias*" too many "and", hard to understand.

L409 "*covariance exists*" it always does, doesn't it? Do you mean that it is large or small?

L410 "*has a negative bias of approximately 7 %,*" hard to compare with the other bias mentionned above which is in kg m-2.

L414 "*R2 = 0.16*" seems like weak correlation. What is the R2 of "*the errors among snow depth and density are uncorrelated*"?

L417 "*Following from the propagation of errors for relative errors in snow depth and density, we estimated the SWE uncertainty to first order (Raleigh & Small, 2017).*" this should be in methods and better explain.

L418 "*The distributed relative SWE uncertainty is presented in Fig. 10 and is negatively correlated with the distributed SWE (R2 = 0.44).*" I understand this as: the larger the SWE, the smaller the error. But I understand that the inverse was true for snow depth and density (L414-415). Sounds inconsistant.

L435 "*Sensitivity analysis*" how does this articulate with the SWE uncertainty in 3.7. I understand that the uncertainty is estimated in two ways: by comparison with the snowpits values and with the sensitivity analysis. This should be clarified in the relevant parts (methods, results) and evaluated.

L440 "*the leading source of error in our density measurement is spatial misalignments*" but further, the impact is estimated to be of 1 kg/m², that is way less than the 15 kg/m² reported or, I guess, the difference with the snow pits.

L441 It is not clear what this cross-correlation is exactly but if it provides the misalignment value, could it not be used to correct it?

L445 What does integration mean?

L447 "*up to 2 %*" use consistent units, kg m-2 in the same sentence.

L455 "*the choice of uncertainties propagated*" it makes it sound like these choices are purely subjectives and not well informed.

L456 "*uncertainty in midwinter tends to reduce at peak SWE*" cite a relevant source for this statement.

L457 "*remarkably difficult*" does the mean density of several snowpits work as well (L475)? If so, the remarkable difficulty is rather to spatialized the density.

L465 "*improved ...importance*" increased?

L470 "*can be treated as random normal*" Say clearly if it has a normal distribution or if it is assumed.

Fig 3, 4, 5, 7: Divergent color map (red, yellow, blue) is not really appropriated for variables which are always positive. It is traditionnaly used for variables centered on zero (yellow).

Fig 3. Could be moved to supplement.

It should be considered to put in a single figure (with adapted colorbar) the snow depth, the density and the swe map. It is hard to identify the patterns similarity as long as they are spreaded over different figures.

Fig 7. This could be moved to supplement and rather show the density map used most in the article (mean of the models).

Figure B1 what are the grey bars? I would move this figure to main manuscript.

Figure C1. Could be merged with Fig 8.

---

## Referee Comment (RC3)

**Review of "Spatially distributed snow depth, bulk density, and snow water equivalent from ground-based and airborne sensor integration at Grand Mesa, Colorado, USA"**

Journal: The Cryosphere Discussions

Authors: Tate G. Meehan, Ahmad Hojatimalekshah, Hans-Peter Marshall, Elias. J. Deeb, Shad O'Neel, Daniel McGrath, Ryan W. Webb, Randall Bonnell, Mark S. Raleigh, Christopher Hiemstra, Kelly Elder

Manuscript ID: tc-2023-141
Review date: November 2023
Reviewer: Kat J. Bormann

**Overview**

In this manuscript, the authors present a method to invert spatially distributed estimates of bulk snow density from the combination of ground-penetrating radar (GPR) retrievals and snow depths generated from airborne lidar. The bulk density map is then combined with a snow depth map at 1 m spatial resolution to calculate spatially distributed SWE for February 1, 2020. The snow density, snow depth and SWE maps are evaluated with field measurements from the NASA SnowEx campaign at Grand Mesa (Colorado, USA) during February 2020. These measurements span forested, exposed, and forest-adjacent areas over relatively flat subalpine terrain during cold and dry winter conditions.

The novel component of this manuscript appears to lie primarily with the generation of spatially distributed bulk snow density estimates from the GPR data using airborne snow depth maps to make the inversion. While algorithms and techniques currently exist for the retrieval of bulk snow density from GPR data (Griessinger et al., 2018; McGrath et al., 2022) the authors present new techniques for automated layer picking and introduce the use of snow depths from airborne lidar. The resulting distributed snow density estimates (after filtering of outliers) show similar accuracies to previous studies (Griessinger et al., 2018) when compared to high-quality snow density measurements, with an RMSE of 27 kg/m$^3$.

The spatial patterns of bulk snow density presented are certainly revealing and will be of great interest to the snow research community. The implications of this variability on SWE retrievals speaks directly to the NASA SnowEx project goals, is an advancement in our understanding, and will be useful for the broader SnowEx community. However, the study misses an opportunity to compare the newly generated SWE maps to existing SWE maps such as those from the Airborne Snow Observatory(ies), which were produced for 1-2 Feb 2020 over the Grand Mesa – using modeled bulk snow density. With this, I think the study falls short of demonstrating improved SWE mapping (using GPR bulk density observations) by failing to compare (and contrast) the newly produced SWE product with existing data ASO that currently represents best practices in SWE mapping (despite the larger spatial resolution of 50m), and potentially other SWE estimates generated as a part of the SnowEx effort. Despite the main thrust of the paper being

the GPR bulk density measurements and methods, I propose that the true scientific advancement of this work lies with such a SWE comparison.

Finally, while I appreciate the breadth of observations and careful use of many sources of field measurements that were used in the study, I found the manuscript narrative to be unfocused at times. Perhaps there is opportunity to reduce scope to include only the pertinent analyses and discussion that support the main purpose of the manuscript.

See attachment for further detail.

*Major/minor comments*

Scope: This manuscript covers a lot of ground, what are the specific questions that are being addressed and how does the analysis presented speak to these questions? I suspect there is information in here that is superfluous. Perhaps reconsider the title of the paper to narrow down and focus the theme.

Comparison with existing techniques and demonstration of improvement: the authors miss a fruitful opportunity to compare their SWE map with other spatially distributed estimates of SWE (such as those prepared explicitly for the NASA SnowEx campaign by the Airborne Snow Observatory(ies), and potentially other SWE maps from instrumentation deployed as a part of the SnowEx 2020 effort. While I understand the spatial resolution of 1m is quite different from the 50m ASO products, the sum of averages should allow a meaningful comparison and contrast and is an obvious omission from the work (in my opinion).
- I realize that with me being a part of the ASO Team, this comment may seem biased. However, I am coming from a place of advancement of the science and understanding – and in my opinion it is common practice to compare new methods with existing methods to demonstrate improvement. Please take my comments with this in mind. I am truly interested in what we can learn from this type of comparison :)

Snow depth map: I am wondering why the authors chose to generate their own snow depth map from the point cloud data using Multiscale Model to Model Cloud Compare (M3C2) rather than just use one of the official SnowEx products from the same lidar data produced by Airborne Snow Observatory(ies). The M3C2 method is really targeted toward benefits over rasterized data in rough and complex terrain, which is far from the flat plateau of the Grand Mesa. This also seems like a missed opportunity to connect the current research/work to existing frameworks within the snow community – for instance, it would be useful to know how these products compare and if M3C2 provides clear benefits (as raster-based subtractions for snow depth are widely used by the community).

Snow density inconsistency: Line 340 "The LiDAR–GPR inferred average snow density shows ... and greater variability in the open terrain than areas sheltered by forest canopies" I would have

expected the opposite, the snow pit measurements do not support this statement (from what I can tell) and, given the MLR/RF/ANN models were trained using the Lidar-GPR densities (with high variability in the exposed areas, it is no surprise that the MLR/RF/ANN models are also disagreeing with the snow pit measurements w.r.t spatial variability of bulk snow density). Perhaps a brief explanation on this inconsistency is required.

Broader appeal: I am wondering if there were other SnowEx locations that had GPR data to replicate this study? (Not required for publication)

**Line edits**

These line edits are mostly questions on the method, meaning or seeking clarification. None of these comments should hinder publication though some of them should probably be addressed.

Line 225 - "We found that errors in the co-registration of these data are the leading source of error in the estimated densities." What is the magnitude of this error?

Line 245 - "At the 1-sigma level, errors of approximately ± 150 kg/m3 can be expected from this sensor integration method". This magnitude of deviation in bulk snow is of course large. I am wondering how the filtering on this would go without "SnowEx level" (high quality and many of) in-situ measurements?

Line 250 – The Grand Mesa is really flat, I'm wondering what magnitude of "geolocation errors in LiDAR-derived snow depths" you are expecting here?

Line 289 - "pixels within a 3 m buffer of LiDAR vegetation greater than 0.5 m height". With snow depths of ~ 1m across the Grand Mesa, are you also including vegetation that was buried?

Line 210: For clarification, the February 2020 flight was actually collected 1-2 Feb (over 2 days)

Line 215:"which agrees well with previous lidar error assessments " and also agrees with the ASO report that was produced for this survey

Line 215: I'm curious about the M3C2 method – were you seeing resulting snow depths of zero depth over the plowed roads?

Line 225: given "spatial registration between the LiDAR and GPR varies on the order of a few metres", and the high spatial variability of snow depth, does it make sense to use 1m snow depths to invert the GPR information? In Section 2.4.3.1 you do an error analysis and peturb TWT and snow depth - of the two perturbations which one injected the most uncertainty into resulting snow density estimate? On this, I really like Figure 3.

Line 255: what was the magnitude of the trends here? "Separate linear trends were identified in the forested and open regions traversed with the GPR"

Line 265: "1 October 2019 through the end of the SnowEx IOP on 12 February 2020" I'm a bit confused as to why you would include 3-12 Feb in your mean wind direction and speed calculations if you are trying to get at snow drifting ON 1-2 Feb. Perhaps I mis-read here.

Line 270: I thought Winstral et al., 2002 was based on the baseline snow off DEM. Why use snow on terrain? "we computed two parameters (maximum upwind slope and wind factor parameters; Winstral et al., 2002) from the 1 February lidar-derived snow surface elevations."

Line 285: Wouldn't some of that 0.5m high vegetation be buried in 1m of snow, and therefore not assert a drift influence? "3 m buffer of LiDAR vegetation greater than 0.5 m height"

Line 330: The Regression models were used to "fill-the-gaps" in the density field provided by the GPR. I'm curious as to what these maps look like.

Line 340: "however within the open and forested domains individually, LiDAR and GPR estimated snow depths are uncorrelated with in situ snow depths (Table 1)." Shouldn't they be correlated somewhat? Why are they uncorrelated with R2 of 0? what do the scatter plots look like?

Line 350: "may explain the weak correlation between estimated density and the in situ measurements" Is the magnitude of the un-correlation around 10%?

Line 405: Consider revising this sentence "which suggests that depth is primary to SWE in this environment"

References

Griessinger, N., Mohr, F. & Jonas, T. Measuring snow ablation rates in alpine terrain with a mobile multioffset ground-penetrating radar system. *Hydrological Processes* **32**, 3272–3282 (2018).

McGrath, D. *et al.* A Time Series of Snow Density and Snow Water Equivalent Observations Derived From the Integration of GPR and UAV SfM Observations. *Frontiers in Remote Sensing* **3**, (2022).

---

## Author Comment (AC1)

**Letter of Response to Reviewers**

Dear Florent Dominé and Reviewers,

First, the authors would like to thank the reviewers for their constructive input that will greatly improve this manuscript. We have carefully mapped out a plan of action to address the comments. In this response letter, we consolidated the reviews into Primary and Editorial Concerns, where primary topics were requested by more than one reviewer. A summary of the comments is listed first in italics, followed by our plan to address the concern.

**Primary Concerns**

*The primary concerns raised by the referees are that the manuscript is unfocused, has a wandering discussion, and is perhaps too rich in methods and analysis throughout.*

In an annotated draft of the preprint, we have color coded our suggested revisions to guide the editor quickly to the proposed changes. Paragraphs requiring simplification/condensing are coded blue, results in the methods section are purple, and methods in the results/discussion sections are pink. These will be moved into the appropriate sections as described in the comments attached to the highlighted text. Figures that are currently referenced in the methods section and convey results will be moved to the results.

The introduction will be significantly shortened to emphasize our work. Editorial emphasis will be placed on the final paragraphs to concisely provide the purpose of the work, downplaying the details of the approach and results. We will clearly state that the purpose of the manuscript is to present a method for efficient snow density estimates over broad spatial scales by integrating GPR and lidar. These observations are used to estimate density and SWE across the entire study domain using regression modeling and supported with evidence-based reasoning. This final point is one which lacks from the body of literature in recent and important works, such as Broxton et al. (2019).

The following Methods subsections will be relocated from the main body of the paper to the appendix or an additional Supplemental Material (at the discretion of the editor according to the format of the journal): 2.4.3.1 *Error analysis of LiDAR – GPR estimated density*; the majority of 2.4.4 *Wind and terrain exposure*. These subsections explain methods which are used to develop our understanding of the spatial data, spatial correlation, and wind processes affecting the observations and models, and are not necessary for many readers. However, information describing our approach of using wind processes as explanatory evidence will remain in the methods section to better frame a key intent of this work.

We will also shorten and focus the discussion, using sub-headings to guide and organize the manuscript. We propose the following discussion sections: deriving density and related uncertainties, distributing observations and related uncertainties, length scales of variability, and physically based explanations of spatial patterns. Presentation of uncertainties early in the discussion should help readers to gain confidence in the work before progressing to more interpretive sections. We can move considerations in regression model development, model transferability, and labor reduction to the Supplemental Material. The purpose-oriented topics – novelty in GPR processing, errors and uncertainties, comparable modeling studies, and explainable spatial patterns – will remain.

Spaceborne: *The link between snow density measurements at the field site and remote sensing is unclear*

We will emphasize the need for distributed density estimates to resolve SWE from space in 1-2 concise sentences. This will be linked to the premise of the SnowEx mission. In short, spatial density estimates are required to convert any snow depth remote sensing product (e.g. lidar or SfM) to SWE, and density is also a major control on snow retrievals from narrow-band SAR.

Point-Cloud Differencing: *Why develop Lidar-based snow depths using point cloud differencing, rather than using existing, raster differenced data products.*

We produced an in-house snow depth product, because to our knowledge no comparable publicly available data product exists. We have since received data privately from Airborne Snow Observatories and agree that to deliver confidence in our approach, it would be a valuable opportunity to compare to other aerial data products, if the ASO data will be public in the near future. We will clarify our approach and the M3C2 jargon.

Co-registration/Geolocation Uncertainty: *Clarify the magnitude of geolocation errors and increase the descriptiveness of the approach.*
Section 3.7 will be subdivided to include: 1. uncertainty analysis of the rasterized SWE product; 2. uncertainty of the derived snow density observations through the combination of radar and lidar information. Much of the reviewer's concerns in subsection 2.4.3.1 can be addressed through reorganization.

Snow Depth Validation: *Provide information and clarification to support data agreement/disagreement.*
A scatter plot which compares the in-situ and lidar derived snow depths in open and forested areas will be added to the appendix/supplement to convey the magnitude of errors which approach the range of snow depths within forested and open areas individually, causing a lack of correlation.

Snow Density Validation: *Data presentation seems contradictory.*
Weak correlation, yet low RMSE of snow density observations – similar to the snow depth validation – will be clarified with additional scatter plots in the appendix/supplement. Points of discussion regarding the uncertainty and known sources of error will be placed in the appropriate sections (i.e., removing results from the methods).

Correlation Length Scale for TWT and SWE: *Clarify the relationship between SWE and TWT, and how this is represented by length-scale analysis.*
The point that TWT and SWE are both dependent on snow depth and density will be clarified and organized into a subsection of the discussion.

**Editorial Concerns**

Most editorial comments will be accepted and noted in the revision through tracked changes except for the few detailed below.

**RC1**

Line 449-450: It is hard to reconcile how the errors in SWE found using this methodology at this field site relate to a global SWE uncertainty target.

*Developing uncertainties at this site, using methodologies which potentially represent a best-case-scenario for broad-scale SWE retrieval, suggest that achieving such a high standard may not be feasible for the global target.*

**RC2**

L24 maybe keep depth and SWE with the same unit, cm?

*The units presented are standard.*

Fig 3, 4, 5, 7: Divergent color map (red, yellow, blue) is not really appropriated for variables which are always positive. It is traditionally used for variables centered on zero (yellow).

*The divergent colormap is useful in this presentation to show the clear differences in snow properties between open and forested areas.*

**RC3**

Line 330: The Regression models were used to "fill-the-gaps" in the density field provided by the GPR. I'm curious as to what these maps look like.

*It is unclear what is being asked by this comment. The regression models are presented in Figure 7, while the Lidar—GPR derived densities are presented in Figure 5.*

The authors miss a fruitful opportunity to compare their SWE map with other spatially distributed estimates of SWE.

*ASO snow depth or SWE products are not within the public domain, such a comparison cannot yet be made openly. Data are available upon request, which the authors have received, and will agree to these comparisons, provided that the data are made public in a timely manner.*

**References**

[revised manuscript text omitted]

---

## Author Response (AR1)

**Letter of Response to Reviewers**

Dear Florent Dominé and Reviewers,

First, the authors would like to thank the reviewers for their constructive input which has greatly improved this manuscript. We have carefully considered our responses and revisions to best address the plan we mapped out initially after review. In this letter we outline the significant changes to the manuscript followed by our detailed responses to the reviewers.

We addressed the primary concerns, that the manuscript was unfocused, had a wandering discussion, and was too rich in methods and analysis throughout.

Our intent was to clarify the set-up of the manuscript which was noticed by all reviewers as muddled. Emphasis in the abstract and introduction was shifted toward how our research provides needed calibration and validation for potential air and spaceborne SWE retrievals and away from general snow remote sensing techniques. The introduction was cut by approximately 25 lines, where statements on snow density modeling and our novel aspects of the work were made concisely and impactfully.

Methods and details had been disseminated throughout the manuscript and were cleaned up in revision. Sections which we thought bogged the reader down were shifted to the appendix. We included a section in the appendix on the LiDAR point-cloud processing applied in our work. We greatly appreciated the attention reviewers gave to this aspect of the work, and we believe it will improve and educate audience understanding. Sections on the machine learning and random gaussian modelling were clarified, and a section describing SWE and uncertainty calculation was added.

It was pointed out that the results sections contained several statements of conjecture which were placed into the discussion or removed. This helped pare down the results which are now simplified and easier to read. A section on the sources of uncertainty was added to replace statements which were mistakenly placed in the methods. A supplement was created to support the various analyses presented in the text. The reviewers pointed out a significant mistake in our presentation of data validation results. We fixed this coding error and have provided scatter plots within the supplement as recommended. We also included comparisons to the ASO snow depth and SWE, as suggested.

The discussion wandered due to a lack of organization. We placed topics within specified subheadings finding many statements which were misplaced or could be improved in their explanation. We intended to cut the superfluous and condensed the discussion by about 100 lines without loss of the important information.

We appreciate the extended time offered to complete this revision, as this effort was a significant overhaul. We hope you all find this revision satisfactory, that no concern was left unaddressed. On behalf of all co-authors, we are pleased to submit this strengthened work to *The Cryosphere*.

Kind Regards,
Tate G. Meehan (corresponding author)

**Responses to Reviewer Comments**

General and specific referee comments were responded to in completion, without exception.

*Our responses are given in italicized text, following the posted comment.*

**RC1**

Line 24: How does this study represent the "next step towards broad-scale SWE retrieval". How does broad-scale SWE retrieval depend on the methods / results shown here?

*Revised to clarify the how, "Spatially continuous snow density and SWE estimated over approximately 16 km2 may serve as necessary calibration and validation for stepping prospective remote sensing techniques toward broad-scale SWE retrieval."*

Introduction: There is dissonance between the initial setup of the study (which is about working toward the improvement of space-based remote sensing and talking about snowpack measurement and mapping at the local scale (e.g. which is measured by GPR, airborne lidar, drones, and field measurements).

*The introduction has been tailored to speak to the methods of data acquisition and modelling, while focusing on how this is relevant for our study.*

Line 34: I think this comment should be revised to "simple spatial interpolation from ground observations is difficult" - things like physiographic variables and past snowpack patterns can be useful for distributing snow measurements.

*Revised as, "Snow varies over such short length scales that spatial interpolation from sparse ground observations is challenging to accurately achieve and validate."*

Line 39: While many of these technologies have, in some circumstances produced fairly accurate maps, there are many caveats and there are also many cases where they fail, or are severely biased.

*Acknowledged*

Line 42: I do not understand what is being said here. The first and second parts of this sentence do not seem to be related.

*Revised via deletion*

Line 81: This paragraph is long and disorganized. The topic sentence is about the spatial patterns of snow density being related to underlying processes, followed by a listing of (sometimes and sometimes not) spaceborne techniques that could be useful for snow density measurement.

*This paragraph has been significantly revised, and broken into various focused paragraphs.*

Line 89: If singling out ground temperature and roughness length, please explain why they are important in this application

*Revised via deletion*

Line 97: Drone based photogrammetry to get snow depth?

*Yes, Clarified*

What is the relationship between GPR measurements to get snow density and space based remote sensing? Can this technique be used to detect snow density using satellite data?

*Revised throughout introduction to clarify our intent with statements such as, "Our research utilizes ground-penetrating radar (GPR), LiDAR, and ML to define an approach to map snow density at resolutions appropriate for air- and space-borne radar remote sensing calibration and validation."*

Line 104: Rather than simply giving a high level overview of your methods, it would be nice to frame this more in terms of research questions, hypotheses, and knowledge gaps.

*Revised introduction throughout to frame the knowledge gaps we address.*

Line 140: Remove the word "whereas"

*Accepted change.*

Line 142-143: "the many forested stands in the survey domain" - this is pretty wordy. Suggest just saying "forested stands"

*Accepted Change*

Why wasn't the same type of instrument used in open and forested areas? How might this have affected results?

*The two systems were required to reach within densely forest areas by ski, and to cover vast areas by snowmachine. I have clarified in the revision that the snowmobile could not travel into densely treed areas. I included a statement within GPR travel-time results (3.2) clarifying, "no systematic bias between the two GPR instruments was found."*

Line 147: Was the same type GPS used on both GPR units (with lower accuracy in the forest)?

*Clarified that a GPS receiver was used on ski, and a GNSS receiver was used on snowmachine.*

Line 160: Were these processing steps the same for both GPR units

*Included a clarification on the processing steps for the skied GPR system.*

Line 213: "Time series" - this is pretty vague…and perhaps not even necessary since you are only using the Feb 1st flight

*Removed time-series terminology.*

Line 215: What is the principle of the Štroner et al method? It is probably better, in the interest of having the work stand on its own, to be somewhat descriptive of all the methods, rather than just referring readers to other papers.

*Additional Details on the point cloud processing has been provided in Appendix B*

Line 230: Change "the" to "in"

*Accepted change*

Line 250-254 - This passage is confusing. How were you able to determine the new random error? It also seems pretty generous to call half the sample (outside the 25th-75th percentiles) outliers.

*See the explanation in the revised Appendix B.2. The percentage of data used to describe outliers is dependant on the degree of misalignment between the radar and lidar sensors. Following work, not presented here, applies the same technique with refinements to the geolocation of data sources which suggests that only bottom and top 10 % of data points need to be considered as outliers going forward. This represents the progression of the technique. See the response to referee 3 for histograms defining the outliers of raw density retrievals.*

Line 270 - Can you be a little more descriptive of how the threshold was determined.

*Revised, "Hourly air temperature data parameterized an empirical relationship to determine the threshold for snow-transportable wind speed (Li and Pomeroy, 1997)."*

Line 293-301: This passage is confusing. Can it be rephrased / reduced?

*This passage has been clarified and reduced within section 2.4.4*

Line 340: Does this mean that the only snow depth differences correctly captured by the lidar was the difference between canopy and open areas. Could this be commented on further in the discussion section (e.g. could it be something related to geolocation errors, differences in the scale of measurements, relatively large lidar errors relative to the variability in either of those environments individually, etc)?

*A mistake was discovered in the validation code, mixing the indices between open and forest areas, which caused this spurious decorrelation. See the response to Referee 2.*

Line 350: Maybe list in the text what this correlation is

*Correlations have been included as suggested, along with a reference to Table 2.b. within the new section 3.8.3*

Line 351-353: "The standard deviation …" - this sentence is awkwardly phrased

Revised as, "The $\rho_{s,Pit}$ and $\rho_{s,LiDAR-GPR}$ data are both normally distributed as evidenced by Z-test (Appendix B) with overlapping standard deviations."

Line 354: Remove "uniquely"

*Accepted change.*

Line 368-369: "the relative length-scales of variability for SWE closely resemble that of TWT and indicate that TWT is a better informer of SWE than either depth or density, independently" - this is confusing. Is there a way to clarify why a smaller difference in length-scale makes TWT a better informer of SWE

*Clarified "… better informer of SWE spatial patterns"*

Line 378: "Average of regression models" - was there just one ANN, RF, and ML model, so that the average is just the average of 3 models, or were there, for instance, an ensemble of ANN models that were averaged together?

*Methods section 2.6.1 was clarified as, "Regression models were trained on the LiDAR–GPR estimated snow density using cross-validation and were applied to the surrounding terrain. Model ensembles were generated and then averaged for both RF and ANN regressions by retraining on random subsets of data. A ML snow density ensemble was composed by averaging the MLR, RF, and ANN outputs. For detail on the parameter estimation and predictor importance see Appendix A."*

Line 405: To get SWE here, do you use average snow density from all pits, as well as the random field field generated from all field density data? It doesn't seem like the SWE comparisons are fair if some methodologies have access to the validation data (e.g. average snow density) and some do not (e.g. regression maps, as far as I can tell).

*Section 2.7 was added to clarify these points.*

Line 408-409: "depth is primary to SWE in this environment" - what does this mean?

*This conjecture has been removed from the article.*

Line 409-410: "Assessed at the average value of SWE for all 96 snow pits" - this is also unclear

*The language has been improved throughout this section 3.7 Spatially distributed snow water equivalent.*

Line 447: 2% - are the units of this supposed to be comparable to the previous number (1 kg/m3)?

*Units have been changed to kg/m³ as suggested.*

Line 449-450: It is hard to reconcile how the errors in SWE found using this methodology at this field site relate to a global SWE uncertainty target.

*Statement has been clarified, "Our uncertainty analysis (Fig. S8) suggests that, even using innovatory measurements from airborne and ground-based sensors, 10 % uncertainty is difficult to achieve."*

Line 539: Could you please define the acronym "FMCW"?

*This paragraph has been removed from the article discussion*

Line 555: Remove comma after SWE. Also, I find this statement confusing.

*Revised as, "Radar travel-time informed dry-snow SWE variability better than either depth or density independently."*

Line 562: Though by comparing SWE estimates at independent snow pits, it seems like you can surmise which method probably performs better**.**

*What is vexing about this entire process is that the true SWE remains unknown. We contend that using LiDAR & GPR, "[has] comparable uncertainty to in situ methods but with spatial continuity at resolutions practical for calibration or validation of air- or space-borne radar remote sensing retrievals of SWE."*

Line 567: Which retrieval methodologies are you referring to here?

*This statement has been removed from the revised conclusions.*

Line 590: I don't think there is a regression learner toolbox. The Regression Learner app is part of the statistics and machine learning toolbox…though neural networks are not part of these tools. How did you implement the neural network models?

*In recent MATLAB versions (I am using 2022a), neural networks are included in this Regression Learner application. The text now clarifies, "application" rather than toolbox.*

**RC2**

L13 A bit surprising to focus here on spaceborne measurement while they are absent of the rest of the article.
*Revised to emphasize snow mass estimations rather than spaceborne remote sensing, "Estimating snow mass in the mountains remains a major challenge for spaceborne remote sensing."*

L13 "enables landscape-scale snow covered area" can "enable" be grammatically used like that?
*This is acceptable grammar; however this has been removed in revision.*

L18 "essential snow physics" is a word missing?
*Revised as, "essential snow research applications".*

L23 "The root-mean-square error between the distributed estimates" A doubt remains whether this RMSE is i) between modeled density and snowpit density measurement or ii) the variability of the various models.
*Clarified, "The root-mean-square error between the distributed estimates and in situ observations…"*

L24 maybe keep depth and SWE with the same unit, cm?
*mm are the standard unit of SWE*

L25 "Wind, terrain, and vegetation interactions display corroborated controls on bulk density that show model and observation agreement." I thought that one conclusion of the article was that snowpit are too sparse to sample the terrain variability.
*Snowpit measurements are too sparse to corroborate this finding. Models referred to here are ML regression densities and wind factor parameters, observations include meteorological data.*

L29 "*declining **of***"?
*Revised as, "snowpacks in the western U.S. declined ~ 20 %..."*

L35 "*from ground observations is not possible*" this should be tempered.

*Text Has been revised as, "…challenging to accurately achieve and validate"*

L38 "*instruments, which*" delete the ","?

*Accepted*

L40 Replace WorldView with "high-resolution satellite stereo images" and maybe keep Marti et al., 2016 and McGrath et al., 2019.

*Accepted*

L42 LiDAR and photogrammetry

*Accepted*

L43 "*whereas*" I do not see opposition between the two parts of the sentence.

*Replaced with "furthermore"*

L51 "*in space more significantly*" => "more in space"

*Accepted*

L54: give the spatial variability scale of snow depth as well

*The spatial variability of snow depth is given in the results of this manuscript.*

L65 Similar, other works looked at converting snow depth to SWE, see for instance and the citations therein:

*Thank you for this information.*

L71 "*more simple*" => "simpler"?

*Accepted*

L116 "*from mean density measured in snow pits with the airborne LiDAR snow depths*"? not clear, please rephrase.

*Revised, "We evaluated the degree of improvement in SWE at the sub-catchment scale by comparing SWE distributed using airborne LiDAR snow depths and density estimated from regression techniques, random-gaussian field synthesis, and the mean density measured in snow pits."*

L131 Cite some of these works.

*Included the following references.*

*Boyd, D. R., Alam, A. M., Kurum, M., Gurbuz, A. C., & Osmanoglu, B. (2022). Preliminary Snow Water Equivalent Retrieval of SnowEX20 Swesarr Data. International Geoscience and Remote Sensing Symposium (IGARSS), 2022-July, 3927–3930. https://doi.org/10.1109/IGARSS46834.2022.9883412*

*Singh, S., Durand, M., Kim, E., Pan, J., Kang, D. H., & Barros, A. P. (2023). A Physical-Statistical Retrieval Framework to Estimate SWE from X and Ku-Band SAR Observations. International Geoscience and Remote Sensing Symposium (IGARSS), 2023-July, 17–20. https://doi.org/10.1109/IGARSS52108.2023.10281838*

L202 Please repeat here the IOP date.

*The dates for the IOP have been reiterated at the beginning of the paragraph, "Snow pit observations and manual depth probe measurements were collected throughout the 27 January – 12 February 2020 IOP…"*

L210 "*the local distance between two point clouds*" This sounds like a significative difference with most snow depth studies (e.g. ASO Painter et al., 2016) in which the snow depth is calculated along the vertical direction as the difference between gridded products. I think it should introduce a discrepancy as well with the snow depth measured in snowpits.

*This technique has been clarified within Appendix B.1*

L219 "*lidar*" written LiDAR elsewhere
*Corrected*

L219 Could the statistics of the elevation difference over stable terrain (snow free, unchanged) be used to estimate the uncertainty?

*This technique is typically accomplished over plowed roads, where snow depth change is expected to be zero. This result has been included in Section 3.8.1.*

L224 "*We applied a k-d tree searcher (Bentley, 1975) to co-register the LiDAR coordinates within a 1 m radius of the GPR TWTs*" I understand co-registration as moving one the elevation dataset for instance with a translation vector (X,Y,Z). Is it the case here? If so, can you cite an article using this k-d tree searcher algorithm? It does not seem very common.

*The kd-tree is simply a fast algorithm for finding coordinate points within a defined radius of a set of query points.*

L226 "*By calculating the maximum cross-correlation lag on continuous segments of transect data*" not clear what this operation is.

*The statement has been clarified as, "By calculating the lag distance of the maximum cross-correlation between GPR TWT and LiDAR snow depth on continuous segments of transect data"*

L239 2.4.3 Error and uncertainty are described at various places of the article (L247, L416- 417, L443). This is confusing. Please reorganize it.

*Uncertainties for the various measurements and results have been placed within a new Section 3.8.*

L249 "*are spatially uncorrelated suggesting that the errors are random and can be treated with filtering.*" = >"are spatially uncorrelated and can be treated with filtering." "filtering" is quite vague, not any filter would be adapted.

*Revised as, "errors are spatially uncorrelated and can be treated with statistical filtering"*

L252 "*we reduced the random error to ± 30 kg/m3 by filtering outliers*." Not clear whether you optimized filtering until this error level was reached or if it is a fortunate consequence.

*Revised as, "Via median filtering and interpolating through outliers error estimates were reduced to 30 kg/m³ (Appendix B.2)."*

L256 When was the last snowfall prior the IOP?

*Certainly, new snowfall would introduce greater discrepancy. Regardless of any new precipitation, it is necessary to consider changes in snowpack height due to densification and wind redistribution, which we have attempted to account for.*

L312-318 This paragraph does not match the title of this section. It could be move in 2.6, in a section gathering all the spatialisation methods used.

*This paragraph has been moved to a new subsection 2.6.2 Random Gaussian Field Model*

L327 What is the spatial resolution of the density maps? All this smoothing must reduce the actual resolution of the maps.

*The density maps are posted with a 1 m pixel size.*

L340 "The in situ snow depth observations (hs,Pit and hs,Probe) compare well with the LiDAR snow depths throughout the entire domain (R2 = 0.61, RMSE = 11 cm, ME = 0 cm), however within the open and forested domains individually, LiDAR and GPR estimated snow depths are uncorrelated with in situ snow depths (Table 1)." This is puzzling. Can a set of points be correlated while two subsets including all the points are not correlated? A simple scatter plot, even in supplement, would help get a better feeling of the data agreement.

*This is an important comment, thank you. A mistake was found in matching the indices of raster pixels to those at probed locations when splitting the data into forested and open areas for validation. Scatter plots have been provided in the supplement, and the tables have been updated with the corrected values.*

L346 "*The GPR data were acquired within a few metres of, but not directly beside the faces of snow pits, which necessitated a radius for pairing observations. The choice of 12.5 m matches that of the filtering during the sensor integration step (Section 2.4.3)*" This should be in method, not results.

*This information has been placed in methods Section 2.2*

L348 "*Measurements accumulated over 12.5 m distance introduce inherent variability on the order of 10 % (Section 3.3), which, along with differences between representative observation scales may explain the weak correlation between estimated density and the in situ measurements.*" This should go in discussion.

*Information regarding the spatial variability has been placed in a new section 3.8.3. Speculation regarding the correlation has been placed in the discussion as recommended.*

L349 "*weak correlation*" sounds contradictory with the 13% RMSE mentioned which sounds good.

*Moving deliberation on weak correlation to the discussion has removed the seemingly contradictory connotation of this statement.*

L353 (and L383) Isn't there strong correlations between Sx, the Wind Factor and the topographic predictors used to estimate the densities? In that case, isn't it expected, by construction, that the densities and the wind proxies will be correlated?

*While true that topographic predictors are used to estimate the density, we see this correlation as well in the GPR-LiDAR measured density. The appearance of snow density features in the machine learning models resembles the upwind slope patterns most closely in the direction of the prevailing wind. This suggests that the terrain features in a particular aspect, here 200-220, are the most causal of the GPR-LiDAR measured patterns and those predicted by the machine learning model ensemble.*

L355 "**suggests** that the method retrieves density patterns which are related to the degree of exposure and shelter due to topography and vegetation" => to discussion

*Revised, as recommended within the discussion.*

L364 "the expected variability among co-located $\rho$ is approximately 2 %," Based on which semi-variogram? Not clear. Again, this sounds more like discussion to me.

*This will be clarified, as either variogram presents a similar result.*

L367 "*to resolve spatial patterns*" => "to resolve spatial patterns at finer scale." There might be spatial pattern of larger scale than 100 m.

*Revised as recommended*

L368 "depth and density **formulate** TWT and SWE," formulate is not clear. By definition, the SWE is the product of depth and density.

*It is too nebulous for the context to clearly describe how TWT is dependent on snow depth and density. The comprehension that both SWE and TWT are related by depth and density is sufficient. The word choice "relate" has replaced "formulate".*

L370 3.4. Please provide metrics comparing the density modeled and measured (mean, median bias, standard deviation, RMSE...).

*Statistical metrics comparing the measured and modeled densities are provided in Table 2.*

L376 "This coincided" what is "this"?

*"This" has been replaced with "The appropriate hyperparametrisation"*

L378 "*as the random field contains little meaningful spatial information*" This is quite expected (idem L388). By construction it is spatially random. I am not sure what is the point of this randomly distributed density. Is it used in other studies?

*The randomly distributed density is presented as an informed benchmark model which we hoped to outperform with the heavy lifting of machine learning. It provides in a visual sense how randomness appears as a snow field, which in turn bolsters the appearance of the spatial patterns supplied by regression modeling.*

L379 "*spreads the strengths*" formulation unclear.

*"Spreads" has been changed to "combines."*

L391 "*The lack of a large-scale topographic trend in density, such as one driven by elevation or aspect, evinces the role of forest vegetation on density.*" is this supported by a figure showing density against elevation or aspect? I don't understand the link between the two parts of the sentence, the elevation/aspect and the forest vegetation.

*Revised, "The role of forest vegetation on snow density is evinced in this topographically simple environment because a large-scale topographic trend, such as one driven by elevation or aspect, is not saturating the signal in density."*

L407 "*with an appropriate correlation length and prior mean and spread but maintain a larger bias*" too many "and", hard to understand.

*Revised, "The ensemble-modelled densities explain more variation in the distributed SWE estimates than those distributed by random gaussian processes."*

L409 "*covariance exists*" it always does, doesn't it? Do you mean that it is large or small?

*Revised as, "The errors among snow depth and density are uncorrelated ($R2 = 0.03$) with negligible covariance."*

L410 "*has a negative bias of approximately 7 %,*" hard to compare with the other bias mentioned above which is in kg m-2.

*Revised to include units of SWE, "bias of -20 mm (7%)"*

L414 "*R2 = 0.16*" seems like weak correlation. What is the R2 of "*the errors among snow depth and density are uncorrelated*"?

*R2 values included, "uncorrelated ($R^2$ = 0.03)"*

L417 "*Following from the propagation of errors for relative errors in snow depth and density, we estimated the SWE uncertainty to first order (Raleigh & Small, 2017).*" this should be in methods and better explain.

*This has been described in a new methods section 2.7*

L418 "*The distributed relative SWE uncertainty is presented in Fig. 10 and is negatively correlated with the distributed SWE (R2 = 0.44).*" I understand this as: the larger the SWE, the smaller the error. But I understand that the inverse was true for snow depth and density (L414-415). Sounds inconsistent.

*The attention to statistical detail in your review has been very helpful. You are correct that these statistics were inconsistent. After doubling back into the analysis code, it was found that both depth and density were negatively correlated with the measurements of 96 the snow pits within the study domain. LiDAR depth errors calculated among the ~30,000 validation depth measurements revealed that the correlation among snow depths was spurious. Snow depth was found to be uncorrelated with the errors ($R^2$ = 0.04). Using simple linear regression, we modelled the snow density errors as a function the machine learning regression ensemble density. Using the RMSE (11 cm) for relative errors in snow depth, linear errors in density, and linear error propagation we estimated the SWE uncertainty to first order (Raleigh & Small, 2017). These errors are presented within the supplement (Fig. S9).*

L435 "*Sensitivity analysis*" how does this articulate with the SWE uncertainty in 3.7. I understand that the uncertainty is estimated in two ways: by comparison with the snowpits values and with the sensitivity analysis. This should be clarified in the relevant parts (methods, results) and evaluated.

*Section 3.8 has been dedicated to uncertainties, and their relevance in our analysis.*

L440 "*the leading source of error in our density measurement is spatial misalignments*" but further, the impact is estimated to be of 1 kg/m2, that is way less than the 15 kg/m2 reported or, I guess, the difference with the snow pits.

*Thank you for the insight. The takeaway is that the roughly 30 kg/$m^3$ error is baked into this analysis, due to the data misalignments. Further perturbations of data alignment led to only a small increase in error.*

L441 It is not clear what this cross-correlation is exactly but if it provides the misalignment value, could it not be used to correct it?

*The value of misalignment is not spatially uniform, attempts were made to correct this, however, I determined it best practice to accept this error and inform the readers of the impact of such misalignments.*

L445 What does integration mean?
*Combining GPR and LiDAR sensor data.*

L447 "*up to 2 %*" use consistent units, kg m-2 in the same sentence.
*Corrected 2 % to 5 kg/m³*

L455 "*the choice of uncertainties propagated*" it makes it sound like these choices are purely subjectives and not well informed.

*Revised as, "Uncertainty propagated"*

L456 "*uncertainty in midwinter tends to reduce at peak SWE*" cite a relevant source for this statement.
*This conjecture has been removed from the article.*

L457 "*remarkably difficult*" does the mean density of several snowpits work as well (L475)? If so, the remarkable difficulty is rather to spatialized the density.

*Subjective language has been removed, "Our findings (Fig. 9) are within the 10 % goal"*

L465 "*improved ...importance*" increased?

*Revised, "increased LiDAR feature predictive power."*

L470 "*can be treated as random normal*" Say clearly if it has a normal distribution or if it is assumed.
*Revised," appears as random normal variable."*

Fig 3, 4, 5, 7: Divergent color map (red, yellow, blue) is not really appropriated for variables which are always positive. It is traditionnaly used for variables centered on zero (yellow).
*Our usage of the divergent colormap is to show anomalous areas, centered around the mean value.*

Fig 3. Could be moved to supplement.
*Placed within Appendix B*

It should be considered to put in a single figure (with adapted colorbar) the snow depth,

the density and the swe map. It is hard to identify the patterns similarity as long as they are spreaded over different figures.

*Thank you for the suggestion, a figure showing depth, density, and SWE side-by-side has been included in the supplement (Fig. S7).*

Fig 7. This could be moved to supplement and rather show the density map used most in the article (mean of the models).

*Accepted, Fig. 6 is now the mean of ML ensemble density.*

Figure B1 what are the grey bars? I would move this figure to main manuscript.

*Bars are the histogram which was then fit with a kernel smoothed density estimator.*

Figure C1. Could be merged with Fig 8.

*Figure C1 has been moved into the Supplement.*

**RC3**

Scope: This manuscript covers a lot of ground, what are the specific questions that are being addressed and how does the analysis presented speak to these questions?

*Our manuscript presents a large portion of material to develop the technology which allows us to ask questions regarding the snow cover. The narrative of the document, perceived as wandering, shows the natural progression of my years of research.*

*First, Large data must be processed and curated. We ask how this can be done most effectively, and we answer that question by developing an automated radar processing technology and utilizing open-source software to reduce an accurate lidar snow depth raster from point cloud data. Then, we must show that LiDAR and GPR can be successfully fused to retrieve snow density. The question we ask, "can this be accomplished," as this technique had not previously been completed at the time I began the research and manuscript.*

*Questions arise in these "engineering" steps, such as is there a signal buried in the noise? We answer that question, yes and yield error bars. When an appropriate filter design is applied, a pattern appears along the radar transects.  We make a significant discovery! What is causing the observed spatial patterns of snow density? What are the correlation lengths of these snow features? And, how can we best distribute this new snow density information?*
*We answered the latter, by providing empirical methods using statistical models and machine learning, to best represent, what we believe are meaningful spatial patterns in the data.*

*Now, we may begin to ask ourselves scientific questions. What physical processes might be driving the patterns we see? How can we infer such physics processes from empirical data? From literature, we pose the wind, terrain, and vegetation hypotheses, and we answer thoroughly, by determining which of our hypothesized terrain and vegetation parameters are most responsible for these patterns and their model dependencies. And we supplement this result to ascertain the impact of wind on snow density patterns, independently. We confirm that wind, in a particular direction, interacts with the terrain and trees of Grand Mesa to explain a significant portion of the pattern variance, both observed by LiDAR—GPR and modeled using terrain features.*

*We learned a lot in this process, it took a lot of heart, and we are wishing to share this entire story with the cryosphere. We intended to have made this story more clear and concise through this revision, despite how long and winding this research has been, and hope this effort shows.*

Comparison with existing techniques and demonstration of improvement: the authors miss a fruitful opportunity to compare their SWE map with other spatially distributed estimates of SWE (such as those prepared explicitly for the NASA SnowEx campaign by the Airborne Snow Observatory(ies), and potentially other SWE maps from instrumentation deployed as a part of the SnowEx 2020 effort. While I understand the spatial resolution of 1m is quite different from the 50m ASO products, the sum of averages should allow a meaningful comparison and contrast and is an obvious omission from the work (in my opinion).

*The authors believe such a data comparison is an important contribution, that had been stifled by data availability. This healthy competition allowed the opportunity to contribute our own lidar snow depth data. The comparison between our data and ASO data has been analyzed in this revision. The results provide insights to the degree of information lost as a function of resolution and lends credence to the accuracy of point could processing.*

Snow depth map: I am wondering why the authors chose to generate their own snow depth map from the point cloud data using Multiscale Model to Model Cloud Compare (M3C2) rather than just use one of the official SnowEx products from the same lidar data produced by Airborne Snow Observatory(ies). The M3C2 method is really targeted toward benefits over rasterized data in rough and complex terrain, which is far from the flat plateau of the Grand Mesa. This also seems like a missed opportunity to connect the current research/work to existing frameworks within the snow community – for instance, it would be useful to know how these products compare and if M3C2 provides clear benefits (as raster-based subtractions for snow depth are widely used by the community).

*Thank you for your suggestion. At the project's initiation, there was no existing snow depth map for February 1, 2020, and the ASO (Airborne Snow Observatory) data had not been published. Within the revised appendix, I have provided additional details about our methodology in response to the previous comment. It's worth noting that the M3C2 method could prove beneficial for flat areas, given its effectiveness on both rough and smooth surfaces.*

Snow density inconsistency: Line 340 "The LiDAR–GPR inferred average snow density shows ... and greater variability in the open terrain than areas sheltered by forest canopies."
I would have expected the opposite, the snow pit measurements do not support this statement (from what I can tell) and, given the MLR/RF/ANN models were trained using the Lidar-GPR densities (with high variability in the exposed areas, it is no surprise that the MLR/RF/ANN models are also disagreeing with the snow pit measurements w.r.t spatial variability of bulk snow density).

Perhaps a brief explanation on this inconsistency is required.

*Snow pit data suggest that the standard deviation of snow density is approximately equal between open (± 19 kg/m$^3$) and forested areas (± 20 kg/m$^3$). As we have stated, the LiDAR—GPR derived snow density snows higher variability in open areas (± 32 kg/m$^3$) than in forested areas*

*(± 19 kg/m³). Modeled results, however variable on model archetype, also tend to show greater variability in the open yet within the same rough order of magnitude as measured in snow pits. We cannot rule out the possibility that snow pit locations incurred a bias due to sampling strategy, nor that the opposite isn't true of the GPR sample locations. We can confidently state, by the amount N fold data observations, that the LiDAR—GPR retrieved density over a greater variety of terrain, and possible variability, within the study domain on Grand Mesa. We also cannot rule out that data inconsistency isn't driven by inherent spatial variability on length scales beneath our threshold of observation due to spatial filtering (10 % variability at 25 m) and that of snow pit measurements (2.5 % nugget variability).*

Broader appeal: I am wondering if there were other SnowEx locations that had GPR data to replicate this study?
*Bonnell et al. (2023) uses similar methods for deriving snow density along GPR transects.*

Line edits

Line 225 - "We found that errors in the co-registration of these data are the leading source of error in the estimated densities." What is the magnitude of this error?
*Errors in retrieved density are initially on the order 150 kg/m³ this error was filtered appropriately which reduced to the random error to approximately 30 kg/m³. This information has been re-presented in Appendix B.2.*

Line 245 - "At the 1-sigma level, errors of approximately ± 150 kg/m3 can be expected from this sensor integration method". This magnitude of deviation in bulk snow is of course large. I am wondering how the filtering on this would go without "SnowEx level" (high quality and many of) in-situ measurements?
*In-situ measurements were used to validate the LiDAR snow depths and derive appropriate error bars of 10 cm therefrom. We also estimated the uncertainty in LiDAR snow depth by maximum standard deviation within the point cloud to raster processing. Uncertainties in GPR travel-times were also derived inherently. Filter design was approached using the range in snow pit densities, however other, more intuitive, choices could have been made. This analysis could be run without the use of in-situ measurements.*

Line 250 – The Grand Mesa is really flat, I'm wondering what magnitude of "geolocation errors in LiDAR-derived snow depths" you are expecting here?
*We expect negligible geolocation errors in the LiDAR product, the intent of this sentence is to convey the fusion of these two data sets, which has been revised. "The combined measurement errors in LiDAR-derived snow depths and GPR TWTs and errors in geolocation upon integrating these data may translate to errors in the retrieved density."*

Line 289 - "pixels within a 3 m buffer of LiDAR vegetation greater than 0.5 m height". With snow depths of ~ 1m across the Grand Mesa, are you also including vegetation that was buried?
*Yes, we include vegetation that is buried by snow cover. We are using the vegetation height segmentation from the September 26, 2016 ASO snow free product. Using a threshold of these*

*data to classify vegetated areas, however, ignores the effect snow cover has on vegetation height.*

Line 210: For clarification, the February 2020 flight was actually collected 1-2 Feb (over 2 days)
*Thank you for the clarification, this has been revised where appropriate throughout.*

Line 215: I'm curious about the M3C2 method – were you seeing resulting snow depths of zero depth over the plowed roads?
*We observed minimal scattered snow depth, ranging between 0 and 5 cm, along the main roads, while higher snow depth (above 1 m) was detected at the edges of the roads (Figure 1). These values were excluded from the final map through masking. This information has been provided in the revision within Section 3.8.1.*

[Figure]

*Figure 1. Snow map of a random road across Grand Mesa. The snow depth values on the road are between 0 and 5 cm.*

Line 225: given "spatial registration between the LiDAR and GPR varies on the order of a few metres", and the high spatial variability of snow depth, does it make sense to use 1m snow depths to invert the GPR information? In Section 2.4.3.1 you do an error analysis and peturb

TWT and snow depth - of the two perturbations which one injected the most uncertainty into resulting snow density estimate? On this, I really like Figure 3.

*For the range in uncertainties expected, we find that snow depth and travel-time both contribute approximately equally to the resulting density. Using a window of 1 m when determining GPR cross-over location errors, substantiates the errors expected by to be encountered during data integration. The use of a 3 m LiDAR snow depth results in a similar density retrieval (Figure 2). The 1 m snow density has a mean and standard deviation of 276 ± 179 kg/m³ where the 3 m snow density is 268 ± 175 kg/m³.*

[Figure]

*Figure 2: The unfiltered retrieved snow density using a 1 m LiDAR raster and a 3 m LiDAR raster.*

Line 255: what was the magnitude of the trends here? "Separate linear trends were identified in the forested and open regions traversed with the GPR"
*The trend has a range of 51 kg/m³ with the minimum value -36 kg/m³ and maximum 15 kg/m³.*

Line 265: "1 October 2019 through the end of the SnowEx IOP on 12 February 2020" I'm a bit confused as to why you would include 3-12 Feb in your mean wind direction and speed calculations if you are trying to get at snow drifting ON 1-2 Feb. Perhaps I mis-read here.

*It seemed more useful for the authors to present the wind data through the duration of the Grand Mesa IOP to the community. GPR data acquired on Feb. 5 was used in this effort. The Median Wind Speed Oct. 1 – Feb.1 is 202 degrees. The median wind speed Oct.1 – Feb. 12 is 200 degrees. The addition of these data does not impact the analysis.*

Line 270: I thought Winstral et al., 2002 was based on the baseline snow off DEM. Why use snow on terrain? "we computed two parameters (maximum upwind slope and wind factor parameters; Winstral et al., 2002) from the 1 February lidar-derived snow surface elevations."

*It seemed intuitive to me to use a snow surface, rather than bare earth, because we are using this information as validation of retrieved snow properties. The time history of snow density patterns we hypothesized to be driven by snow surface expression after a period of snow accumulation. The difference in results is negligible.*

Line 285: Wouldn't some of that 0.5m high vegetation be buried in 1m of snow, and therefore not assert a drift influence? "3 m buffer of LiDAR vegetation greater than 0.5 m height"

*This is good reasoning to use the snow surface elevation model in these calculations.*

Line 330: The Regression models were used to "fill-the-gaps" in the density field provided by the GPR. I'm curious as to what these maps look like.

*It is unclear what is being asked by this comment. The regression modelled density is presented in Figure 6, while the Lidar—GPR derived densities are presented in Figure 5.*

Line 340: "however within the open and forested domains individually, LiDAR and GPR estimated snow depths are uncorrelated with in situ snow depths (Table 1)." Shouldn't they be correlated somewhat? Why are they uncorrelated with R2 of 0? what do the scatter plots look like?
*This was a helpful comment which drew our attention to a mistake in the tracking of indices between validation data in forest and open areas. The text has been corrected in Section 3.2 and Table 1 which overviews these data.*

Line 350: "may explain the weak correlation between estimated density and the in situ measurements" Is the magnitude of the un-correlation around 10%?
*Yes, for an average snow density of 275 kg/m$^3$, a 10 % error would be 27.5 kg/m$^3$ which agrees exactly with the RMSE of the snow pit validation.*

Line 405: Consider revising this sentence "which suggests that depth is primary to SWE in this environment"
*Revised. "Using an average measured value performs slightly better."*

---

## Referee Report (RR1)

**Review of Spatially distributed snow depth, bulk density, and snow water equivalent from ground-based and airborne sensor integration at Grand Mesa, Colorado, USA**
**Meehan et al., The Cryosphere  Discussion**

César Deschamps-Berger

The authors improved significantly the article by reorganizing and reducing the main body of the text. The work gained in clarity and the main results are better highlighted. The article remains rich in methods and some analysis are still a bit hard to follow but overall, I only suggest to take into account the following minor comments. For futur works, representation of the various estimations of SWE, snow depth and density could be clearly represented with bar plots and whiskers for uncertainty.

**Minor comments**

**L13** for spaceborne remote sensing => « with remote sensing methods », the article has little to see with spaceborn methods.

**L28** Maybe add a more general sentence about climate change impact on snowpack at global scale before zooming on the western US ?

**L43** « light detection and ranging (LiDAR; e.g., Deschamps-Berger et al., 2023; Hu et al., 2021) aboard ICESat-2 (Abdalati et al., 2010). ». I suggest to also cite Treichler et al. (2017) with ICESat and Besso et al. (2024) with ICESat-2. Abdalati et al. (2010) is a bit outdated, it was published 8 years before the launch of ICESat-2. I do not think it is necessary to cite an article for ICESat-2 but in that case, Magruder et al. (2021) could be an option :

« light detection and ranging (LiDAR) with ICESat (Treichler et al., 2017) and ICESat-2 (e.g., Hu et al., 2021 ; Deschamps-Berger et al., 2023; Besso et al., 2024).»

Besso, H., Shean, D., Lundquist, J., Mountain snow depth retrievals from customized processing of ICESat-2 satellite laser altimetry, *Remote Sens. Environ.*, Volume 300, 113843,ISSN 0034-4257,https://doi.org/10.1016/j.rse.2023.113843, 2024.

Magruder, L., Neumann, T., & Kurtz, N. ICESat-2 early mission synopsis and observatory performance. *Earth and Space Science*, 8, e2020EA001555. https://doi.org/10.1029/2020EA001555, 2021.

Treichler, D. and Kääb, A.: Snow depth from ICESat laser altimetry– A test study in southern Norway, *Remote Sens. Environ.*, 191, 389–401, https://doi.org/10.1016/j.rse.2017.01.022, 2017.

**L57** « empirical models spatially distribute density in SWE estimates » => « empirical models are used to spatially distribute density in SWE estimates » ?

**L65** « Machine learning (ML)... » Is ML different from empirical models ? The same Broxton et al. (2019) citation is used L57. It makes this last sentence a bit confusing.

**L65-67** Further, how different is « verification » from « validation » ? Verification sounds vague and makes the whole sentence circular (validation is lacking requiring validation to gain confidence).

**L70** « simpler » => « simple » ? or simpler than what ?

**L80** « require appropriate constraints » quite vague, please precise.

**L90** «Our work leverages **airborne** LiDAR » ?

**L95** « highlights interactions between snow [...] on the densification process » interactions on ?

**L97** « to assimilate parameterizations » assimilation data is the usual term, what are you refering to here?

**L155** Eqn 1. It not clear how $C_{HH-HV}$ would be calculated from this equation. Is it by setting i=1 and i=2, respectively in the first and second sum ?

**L160** Eqn 2. Does $C_{HH}$−HV depend on *t* ? Add *max(C)*. The max is calculated within the window or all the data ? $C_{HH}$−HV and $C$(t) seem interchangably used for coherence (L155, L158) and normalized coherence (L160). Please clarify the notation.

**L190** « were resampled to the 1 m resolution » using nearest-neighboor algorithm ? Precise.

**L197** « to co-register the LiDAR » from your answer, I understand that you do not shift or translate the LiDAR data but rather associate LiDAR points with snowpits. Could « pair », « identify », « match » be more appropriate than « co-register » ?

**L243** : « by retraining on random subsets of data. » not clear at all.

**L260** « upscaled » how ? Using what algorithm ?

**L262** «the RMSE (11 cm) was used to estimate the random error » RMSE is impacted by systematic errors (i.e. bias), thus it should not be used to estimate the random error.

**L302** « Using supervised ML regression**,** » add coma ?

**L20** « drives **SWE** spatial patterns » ?

**L388-390** « that are on the scale of the 1 m resolution data products. » I am a bit doubtfull of that. The density models were trained on raster variables smoothed at 5 m and 25 m resolution. Thus a 1 m shift is small compared to the actual resolution of the densities and should have little impact. Especially taking into account the little variability of density at this scale (Fig 5.). Following your answer to my previous similar comment : how do you understand the fact that « f*urther perturbations of data alignment led to* » a much smaller error (1 kg m-3) than the error mostly attributed to misalignment (30 kg m-3)?

**L403**. Maybe comment on the fact that Yildiz et al. (2021) had a smaller study site which limited the lag considered to a maximum of ~50 m ?

**L435** « To capture the range of processes (i.e., elevation, slope, aspect, and forest attributes) » these are not processes.

**L468** « is...was » unify tenses ?

**L483** « a SnowEx pit in two hours » give an estimate of the depth of the pit as it is has a major effect on required time.

**L501** « that snow pit observations are independent and unable to resolve spatial patterns < 150 m scale » this a result of the sampling strategy, not of the snow pit approach in itself.

**Fig 2**. State in the legend that the colorbar is centered on the mean value. Idem in similar figures.

**Fig 5**. Interesting, more variability in depth and SWE in forest compared to open, but less in density. Could be worth commenting in 3.4 ? Could it result from relative importance of wind transport, canopy interception… ?

**Fig. B2**. The yellow histogramm is hardly visible. Maybe make it darker ?

**Fig S8**. Did you filter out negative values ? If so, state it in the methods, if not use a colorscale allowing negative values.

---

## Author Response (AR2)

**Responses to Editor & Reviewer Comments**

General and specific referee comments were responded to in completion, without exception.

*Our responses are given in italicized text, following the posted comment.*

**Editor Comments**

Line 59. "grain-bond characteristics" may not be clear to all readers. How about "snow type" or "snow type e.g. depth hoar vs rounded-grain snow", or any other option you would propose.

*Revised as, "snow type characteristics (e.g., faceted crystals versus rounded-grain snow)"*

Lines 217-220. I had to read this sentence 3 times to really understand it. How about breaking it into 2 shorter sentences? In particular, the construction ", we suggest," is far from ideal.

*This sentence has been removed, as this topic is provided in the discussion.*

**Referee Comments**

**L13** for spaceborne remote sensing => « with remote sensing methods », the article has little to see with spaceborn methods.

*Accepted*

**L28** Maybe add a more general sentence about climate change impact on snowpack at global scale before zooming on the western US ?

*The authors have decided to maintain the focus of the article on the western US and how this research pertains to our study of Grand Mesa, Colorado.*

**L43** light detection and ranging (LiDAR; e.g., Deschamps-Berger et al., 2023; Hu et al.,2021) aboard ICESat-2 (Abdalati et al., 2010).

*Revised as recommended, "light detection and ranging (LiDAR) with ICESat (Treichler and Kääb, 2017) and ICESat-2 (Hu et al., 2021; Deschamps-Berger et al., 2023; Besso et al., 2024)."*

**L57** empirical models spatially distribute density in SWE estimates => empirical models are used to spatially distribute density in SWE estimates

*Revised as recommended.*

**L65** « Machine learning (ML)... » Is ML different from empirical models ? The same Broxton et al. (2019) citation is used L57. It makes this last sentence a bit confusing.

*Elder et al., 1998, Wetlaufer et al., 2016, and Broxton et al., 2019 now reference the ML approaches on L66 and have been removed from L57.*

**L65-67** Further, how different is « verification » from « validation » ? Verification sounds vague and makes the whole sentence circular (validation is lacking requiring validation to gain confidence).

*Revised as, "… are often distributed over vast areas with little validation, or consideration to the underlying physical processes, required to gain an acceptable level of model confidence."*

**L70** « simpler » => « simple » ? or simpler than what ?

*Simpler than dynamic compaction schemes. The word simpler has been removed, as it is subjective.*

**L80** « require appropriate constraints » quite vague, please precise.

*Revised as, "Yet, many radar remote sensing retrievals require constraints on the snow depth, density, stratigraphy, and microstructure to be presently reliable (Tsang et al., 2022)."*

**L90** «Our work leverages **airborne** LiDAR » ?

*The word airborne has been added to the sentence as recommended.*

**L95** « highlights interactions between snow [...] on the densification process » interactions on ?

*"On" has been changed to "in", as, "…highlights interactions between snow, terrain, vegetation, and wind in the densification process…"*

**L97** « to assimilate parameterizations » assimilation data is the usual term, what are you refering to here?

*Revised as, "Our work addresses the need for high accuracy, distributed density measurements as assimilation data for parameterizations of snow densification…"*

**L155** Eqn 1. It not clear how $CHH-HV$ would be calculated from this equation. Is it by setting i=1 and i=2, respectively in the first and second sum ?

*The Equation has been clarified by correcting the notation of the $j^{th}$ iteration in the summation as $C(t) = \frac{1}{2} \sum_{j=t-N/2}^{t+N/2} \left\{ \left[ \sum_{i=1}^{M} S_{i,j} \right]^2 - \sum_{i=1}^{M} S_{i,j}^2 \right\}$,*

**L160** Eqn 2. Does $CHH-HV$ depend on $t$ ? Add *max(C)*. The max is calculated within the window or all the data ? $CHH-HV$ and $C$(t) seem interchangably used for coherence

(L155, L158) and normalized coherence (L160). Please clarify the notation.
*Corrected*

**L190** « were resampled to the 1 m resolution » using nearest-neighbour algorithm ?
Precise
*Revised as, "…bare-earth and vegetation data products were resampled using the nearest-neighbour approximation…"*

**L197** « to co-register the LiDAR » from your answer, I understand that you do not shift or translate the LiDAR data but rather associate LiDAR points with snowpits. Could « pair », « identify », « match » be more appropriate than « co-register » ?

*The word "find" has replaced "co-register"*

**L243** : « by retraining on random subsets of data. » not clear at all.

*Revised as, "By retraining the model architectures on random subsets of data, 50 model ensembles were generated and then averaged for both RF and ANN regressions."*

**L260** « upscaled » how ? Using what algorithm ?

*Revised as, "We upscaled $b_{s,LiDAR-\overline{Ens}}$ to 50 m resolution using nearest-neighbour approximation for comparison with the 50 m ASO SWE."*

**L262** «the RMSE (11 cm) was used to estimate the random error » RMSE is impacted by systematic errors (i.e. bias), thus it should not be used to estimate the random error.

*Please note that our LiDAR snow depth product was validated and expresses no systematic biases.*

**L302** « Using supervised ML regression**,** » add coma ?

*Accepted*

**L320** « drives **SWE** spatial patterns » ?

*Accepted*

**L388-390** « that are on the scale of the 1 m resolution data products. » I am a bit doubtfull of that. The density models were trained on raster variables smoothed at 5 m and 25 m resolution. Thus a 1 m shift is small compared to the actual resolution of the densities and should have little impact. Especially taking into account the little variability of density at this scale (Fig 5.). Following your answer to my previous similar comment : how do you understand the fact that « *further perturbations of data alignment led to »* a much smaller error (1 kg m-3) than the error mostly attributed to misalignment (30 kg m-3)?

*Errors caused by further perturbations of data alignment are insensitive to additional perturbation, as you have alluded to, by our processes of smoothing data and outlier filtering. In practice, we find that within a 1 m radius of probed depth validation measurements approximately 10 cm of error in snow depth and approximately 1 ns of TWT error from cross-over analysis is expected. We show how these measurement errors contribute to errors of up to 150 kg/m^3 in the density retrieval. A small perturbation in space may in fact produce a large error. Without errors existing between the registration of these data, and if you comprehend that the measurements of both LiDAR snow depth and GPR travel-time are repeatable in space, our method of retrieval is straightforward and depth and TWT will agree with snow density.*

*Retrieval error, as just described, is challenging to disentangle from validation errors and inherent spatial variability. Within a snow pit measurement of density, it is very common to make adjacent measurements which vary by more than 25 kg/m^3. We provide, that on average across all snow pits the sample variability among adjacent data is 2.5 %. We also find this to be the inherent, nugget variability of our retrieval algorithm. Which suggests that our retrieval is sensitive to real variability and at the appropriate length scales.*

**L403**. Maybe comment on the fact that Yildiz et al. (2021) had a smaller study site which limited the lag considered to a maximum of ~50 m ?

*Revised as, "These findings differ from a previous variogram analysis that found correlation lengths for snow density of less than 10 m at a smaller study site which limited the maximum lag separation to approximately 50 m (Yildiz et al., 2021)."*

**L435** « To capture the range of processes (i.e., elevation, slope, aspect, and forest attributes) » these are not processes.

*"processes" has been changed to "terrain features"*

**L468** « is…was » unify tenses ?

*Corrected*

**L483** « a SnowEx pit in two hours » give an estimate of the depth of the pit as it is has a major effect on required time.

*Revised to clarify a 1 m deep snow pit. "For example, a team of two can fully sample a one-metre-deep SnowEx pit in two hours…"*

**L501** « that snow pit observations are independent and unable to resolve spatial patterns < 150 m scale » this a result of the sampling strategy, not of the snow pit approach in itself.

*Revised for clarity as, "…snow pit observations following the SnowEx 2020 Grand Mesa IOP sampling strategy are independent and unable to resolve spatial patterns < 150 m scale."*

**Fig 2**. State in the legend that the colorbar is centered on the mean value. Idem in similar figures.

*Corrected*

**Fig 5**. Interesting, more variability in depth and SWE in forest compared to open, but less in density. Could be worth commenting in 3.4 ? Could it result from relative importance of wind transport, canopy interception… ?

**Fig. B2**. The yellow histogramm is hardly visible. Maybe make it darker ?

*Corrected*

**Fig S8**. Did you filter out negative values ? If so, state it in the methods, if not use a colorscale allowing negative values.

*Revised the Text in Section 2.4.2, "Then, we applied the point cloud differencing method to estimate snow depth on a 1 m grid (Appendix B.1). Negative snow depth values were filtered as no data values."*

---

## Author Response (AR3)

Dear Editorial Staff of *The Cryosphere*,

We are pleased and grateful for our manuscript to have been accepted for publication within your esteemed snow science journal. We have worked hard to correct our article throughout the review process, and upon this final iteration we have resolved the noted issues within the manuscript reference list. To correct this, I have checked to ensure that each intext reference is cited within the bibliography. To ensure that we have the correct format, I generated the bibliography in Latex using the Copernicus style file. However, I noticed that the journal names within the reference list were not abbreviated. Care was taken to adjust the journal names to desired abbreviation. Also included is the .bib file containing the references cited within our text, if helpful.

Thank you again for your patience and helpful guidance through this process. Do not hesitate to reach out if for any reason the manuscript has been prepared unsatisfactorily.

Kind Regards,

Tate G. Meehan